# Carbon costs and benefits of Indonesian rainforest conversion to plantations

Thomas Guillaume[1,2,3], Martyna M. Kotowska[4], Dietrich Hertel[4], Alexander Knohl[5,6], Valentyna Krashevska[7], Kukuh Murtilaksono[8], Stefan Scheu[6,7] & Yakov Kuzyakov[1,9]

Land-use intensification in the tropics plays an important role in meeting global demand for agricultural commodities but generates high environmental costs. Here, we synthesize the impacts of rainforest conversion to tree plantations of increasing management intensity on carbon stocks and dynamics. Rainforests in Sumatra converted to jungle rubber, rubber, and oil palm monocultures lost 116 Mg C ha$^{-1}$, 159 Mg C ha$^{-1}$, and 174 Mg C ha$^{-1}$, respectively. Up to 21% of these carbon losses originated from belowground pools, where soil organic matter still decreases a decade after conversion. Oil palm cultivation leads to the highest carbon losses but it is the most efficient land use, providing the lowest ratio between ecosystem carbon storage loss or net primary production (NPP) decrease and yield. The imbalanced sharing of NPP between short-term human needs and maintenance of long-term ecosystem functions could compromise the ability of plantations to provide ecosystem services regulating climate, soil fertility, water, and nutrient cycles.

[1] Soil Science of Temperate Ecosystems, University of Göttingen, Büsgenweg 2, Göttingen 37077, Germany. [2] School of Architecture, Civil and Environmental Engineering (ENAC), Ecole Polytechnique Fédérale de Lausanne (EPFL), Ecological Systems Laboratory (ECOS), Station 2, Lausanne 1015, Switzerland. [3] Swiss Federal Institute for Forest, Snow and Landscape Research (WSL), Site Lausanne, Station 2, Lausanne 1015, Switzerland. [4] Albrecht-von-Haller Institute for Plant Sciences, University of Göttingen, Untere Karspüle 2, Göttingen 37073, Germany. [5] Bioclimatology, University of Göttingen, Büsgenweg 2, Göttingen 37077, Germany. [6] Center of Biodiversity and Sustainable Land Use, University of Göttingen, Von-Siebold-Str. 8, Göttingen 37075, Germany. [7] J. F. Blumenbach Institute of Zoology and Anthropology, University of Göttingen, Untere Karspüle 2, Göttingen 37073, Germany. [8] Department of Soil Science and Land Resources, Bogor Agricultural University, Jl. Meranti, Darmaga Campus, Bogor 16680, Indonesia. [9] Department of Agricultural Soil Science, University of Göttingen, Büsgenweg 2, Göttingen 37077, Germany. Correspondence and requests for materials should be addressed to T.G. (email: thomas.guillaume@epfl.ch)

Global demand for agricultural products calls for an increase in agricultural productivity[1,2]. Tropical regions are at the forefront of agricultural expansion and intensification to meet this demand due to rapid economic development, substantial potential to achieve high yield, and availability of unexploited land[3]. Nonetheless, agricultural expansion and intensification in the tropics are associated with substantial environmental impacts, such as increased greenhouse gas emissions, biodiversity loss, and soil degradation, and is expected to aggravate conflicts between nature conservation and food production[4]. Consequently, agricultural eco-efficiency needs to be enhanced, i.e., more output must be produced to satisfy human needs with less consumption of natural resources[5].

Agricultural land in the tropics expanded by 100 million ha between 1980 and 2000, mostly at the expense of natural, sustainably used, or disturbed tropical forests[6,7]. Oil palm (*Elaeis guineensis*) and rubber (*Hevea brasiliensis*) cultivation played a significant role in this expansion and reached 30 million ha worldwide in 2014, with 37% of the total area located in Indonesia[8]. The large past and forecasted expansion of South-East Asia's oil palm and rubber plantations illustrates how emerging tropical countries rely on these perennial crops to increase their economic welfare[9–11]. The downside of this development is an increase in deforestation to such an extent that Indonesia became the country with the highest deforestation rate in 2012[12].

The conversion of natural ecosystems to agroecosystems in the tropics is associated with lower carbon (C) eco-efficiency in terms of C storage decrease per unit of harvested biomass than in temperate ecosystems. This is predominantly due to the large amounts of C stored in trees in tropical forests and lower average yields[7]. However, intensive land uses such as oil palm and rubber cultivation that rely on fertilizers and phytosanitary inputs are becoming more frequent[1], replacing extensive and low-input land-use types such as rubber agroforest[13]. While yield increase enables more agricultural commodities to be produced per surface area, intensive plantations require higher consumption of natural resources such as water and nutrients, and deliver less regulating ecosystem services than natural ecosystems or less intensive land-use types[14]. Additionally, a high proportion of harvested biomass reduces the C and energy available for heterotrophic organisms in the plantation's food web, thereby limiting ecosystem services supporting agricultural production[15].

Quantifying the gains and costs in C storage and productivity after rainforest conversion to agricultural land is fundamental to advise tropical countries in their development policies. The belowground C dynamics under perennial crops remains largely unknown and contradicting results emerge from literature[16]. For instance, soil organic carbon (SOC) stocks after rainforest conversion to oil palm have been reported to decrease[17–20], remain constant[21,22], or even increase[23]. Due to the lack of available or consistent data, the International Panel on Climate Change (IPCC) Guidelines for National Greenhouse Gas Inventories do not provide default values for belowground C stocks in perennial crops, and the Tier 1 methodology accounts only for aboveground C stock changes when rainforests on mineral soils are converted to tree plantations[24]. Similarly, methodologies developed to certify the sustainability of new plantations on mineral soil (e.g., Roundtable on Sustainable Palm Oil (RSPO) or High Carbon Stocks (HSC) approach) do not account for changes in SOC, and in some cases, in belowground plant biomass[25,26].

Unfortunately, assessments of the impact of natural ecosystem conversion to agroecosystems including all C pools and fluxes are scarce because they require a combination of various scientific disciplines and are difficult to conduct by a single research group. The multidisciplinary project, Ecological and Socioeconomic Functions of Tropical Lowland Rainforest Transformation

Systems, offered an exceptional opportunity to combine interdisciplinary data to address the broad range of impacts of tropical land-use change on all major components of the C cycle within plantation boundaries. More specifically, our aim was to quantify the impacts of rainforest conversion to plantations of increasing agricultural intensity in terms of harvested biomass, fertilizers, and herbicide application, i.e., rubber agroforests (jungle rubber) with low yield and no fertilizer or herbicide applications, rubber monocultures, and oil palm monocultures both with high yield, fertilizer, and herbicides applications[14]. We synthesized data from aboveground and belowground C pools down to 50 cm depth, as well as C fluxes published by research groups working on the same plots in the Jambi province in Indonesia—as a typical example of land-use intensification in an emerging tropical country[12]. Carbon stocks in each land-use type were measured once in each of the eight replicate plots (aboveground biomass, dead wood, litter, coarse roots biomass, living and dead fine roots biomass, SOC) whereas C fluxes (net primary production, soil $CO_2$ efflux, and net $CH_4$ uptake) and litter decomposition were monitored over 1 year with monthly measurements for gases. Aboveground and coarse roots biomass data, as well as wood and coarse roots production, were updated with recently published allometric equations[27,28]. First, our aim was to quantify total C losses after rainforest conversion to jungle rubber, rubber, and oil palm monocultures. Second, we identified which above- and below-ground pools are most sensitive to conversion. Third, we investigated net ecosystem productivity (NEP), ecosystem C storage, and SOC dynamics using the balance between biomass C inputs and soil $CO_2$ efflux. Finally, we put the yield increase resulting from land-use intensification into perspective with the reduction in ecosystem C storage and biomass production. We find that rainforest conversion to oil palm plantations leads to the highest ecosystem C storage loss because of the shorter rotation time of oil palm plantations compared to rubber. Even though most C losses occur aboveground, significant C amounts were lost belowground. Soil organic C stocks are still not at equilibrium more than a decade after conversion. Therefore, losses from mineral soils should be accounted for when rainforests are converted to perennial crops. When C losses and the reduction of C available for heterotrophic organisms are put into perspective with the harvested biomass, oil palm is the most efficient land use. However, the slowdown of soil organic matter cycling (production and decomposition) in monocultures questions the sustainability of these land-use types.

## Results

**Carbon stocks.** Conversion of tropical rainforest to tree plantations greatly reduced C storage in the investigated ecosystems (Fig. 1, Supplementary Table 1). As expected, the largest C losses among all C pools occurred in aboveground biomass (AGB), reaching 102–131 Mg C ha$^{-1}$ depending on the plantation type and corresponding to 73–88% of total C losses. When all aboveground and belowground (down to 50 cm depth) C pools are considered, C losses together at the time of measurement reached $116 \pm 16$ Mg C ha$^{-1}$ (mean ± standard error, threshold for significance: $p$-values < 0.05), $159 \pm 17$ Mg C ha$^{-1}$, and $174 \pm 13$ Mg C ha$^{-1}$ for jungle rubber, rubber, and oil palm plantations, respectively (Table 1). Hence, up to 61% of the $284 \pm 12$ Mg C ha$^{-1}$ stored in the undisturbed ecosystem was lost in managed systems. The estimated AGB stocks presented in this synthesis and updated from published data[29] using more recent allometric equations[27,28] did not significantly change for rainforest (+2.7%), jungle rubber (+5.5%), and rubber (+0.3%) plantations as compared to previously published AGB stocks. The updated oil palm AGB stocks, however, increased by 86%. Accordingly, even

though oil palm had the lowest C storage over the investigated systems (Fig. 1), differences were not significant with rubber monocultures. Nonetheless, oil palm plantation was the only agroecosystem that had a significant decrease in total

belowground C storage down to 50 cm depth (sum of coarse roots, fine roots, fine roots necromass, and soil organic C) as compared to rainforest (Supplementary Table 1). SOC—the main belowground C stock—overall was not affected by rainforest conversion to plantations over 7–17 years, yet plantations in the loamy Acrisol region experienced losses of up to 15 Mg C ha$^{-1}$ of SOC in the topsoil compared to rainforest when analyzed separately[18]. The largest belowground losses occurred in roots biomass (16–28 Mg C ha$^{-1}$ lost depending on land use), with more than 97% resulting from coarse root biomass loss. Despite a strong decrease in total biomass, oil palm tended to have higher fine root biomass and had higher fine root necromass than any other land-use type (Fig. 2). Dead biomass (dead wood, litter, and fine root necromass) represented only 5% of total C stocks in rainforest and jungle rubber; and this fraction was further reduced because dead wood is absent in monoculture plantations. Considering the soil C stocks down to 50 cm depth, rainforest stored 1.6 times more C aboveground than belowground. However, because SOC stocks respond more slowly to land use-change compared to plant biomass stocks, the majority of C remained stored belowground in plantations.

**Carbon fluxes.** Carbon inputs into the ecosystems were affected in opposite directions when rainforest was converted to rubber or to oil palm plantations[29]. Total net primary production (NPP$_{tot}$; wood, leaves, branches, reproductive organs, coarse and fine roots, and harvested biomass) was strongly reduced in rubber monocultures ($-33 \pm 7\%$) as compared to rainforest, but it did not change significantly in extensive rubber plantations ($-18 \pm 6\%$), whereas NPP$_{tot}$ increased by $47 \pm 9\%$ in oil palm plantations (Fig. 1, Supplementary Table 1). More than half ($58 \pm 3\%$) of the biomass produced per year in oil palm plantations was harvested and thus exported from the ecosystem in the form of fresh fruit bunches. The proportion of biomass production harvested in the form of latex was much lower in rubber monocultures ($25 \pm 5\%$) and jungle rubber ($5 \pm 1\%$). Accordingly, the biomass produced and remaining in the ecosystem (NPP$_{eco}$; wood, leaves, branches, reproductive organs, and coarse and fine roots) was reduced by $51 \pm 6\%$ in rubber and $39 \pm 7\%$ in oil palm plantations, but only by $21 \pm 7\%$ in jungle rubber as compared to rainforest (Fig. 1). The biomass production of each vegetative parts followed the same trends, i.e., rainforest > extensive rubber > rubber ≈ oil palm. The main difference in C allocation between rainforest and monocultures was that rubber trees, and especially oil palms, invested more C into fine roots than into coarse roots (Fig. 2).

Another main difference between rainforest or jungle rubber with monocultures is that the C accumulated in the wood biomass in monocultures is not available for the decomposer food web due to the absence of tree turnover in monocultures. Hence, the maximum amount of fresh C available for heterotrophs

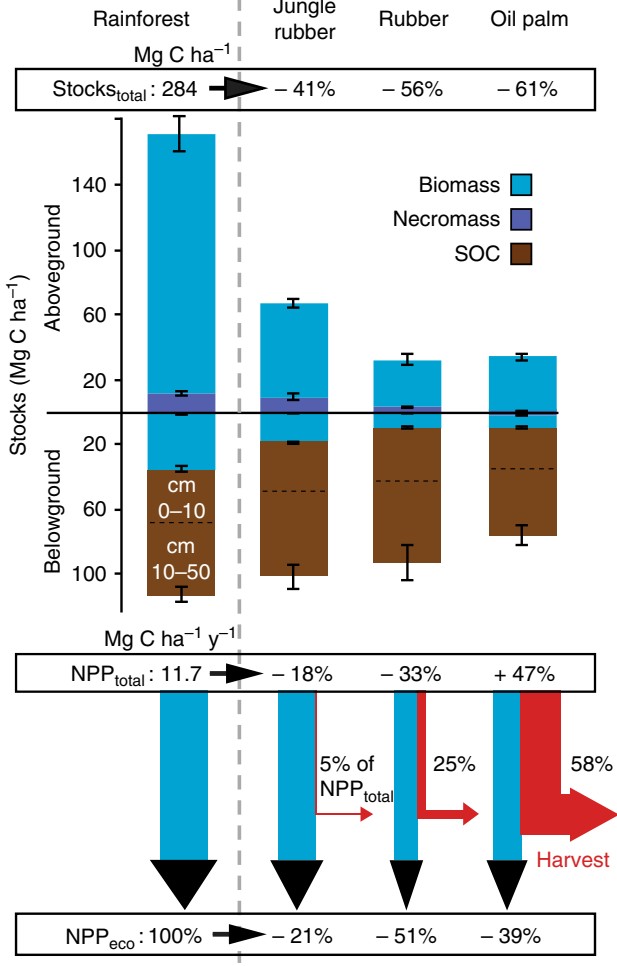

**Fig. 1** Carbon stocks and net primary production of four land-use types. Relative changes (%) compared to rainforest are presented. Belowground biomass is the sum of coarse and fine root carbon (C) stocks. Aboveground necromass is the sum of dead wood and litter stocks. Belowground necromass corresponds to root necromass stocks. The harvested biomass (red arrows) is presented as a percentage of total net primary production (NPP$_{total}$) of the plantation. NPP$_{eco}$ is NPP$_{total}$ minus harvested biomass, i.e., NPP remaining in the ecosystem (wood, litterfall, coarse, and fine roots). Error bars represent the standard error of the mean (SE)

**Table 1 Gain and cost of land-use change**

| Land uses | Total C losses Mg C ha$^{-1}$ | Net biomass C uptake Mg C ha$^{-1}$ y$^{-1}$ | Time-averaged biomass C stocks Mg C ha$^{-1}$ | C efficiency y$^{-1}$ | NPP tradeoff |
|---|---|---|---|---|---|
| Jungle rubber | 116 ± 16[a] | n.d. | n.d. | 114–738[b] | 0.0–0.6[b] |
| Rubber | 159 ± 17 (134 ± 18)[c] | 3.1 ± 0.2 (2.3 ± 0.2)[d] | 62 ± 4 (47 ± 3)[e] | 30–311 | 0.2–0.7 |
| Oil palm | 174 ± 13 (173 ± 14) | 3.3 ± 0.1 (2.7 ± 0.1) | 41 ± 2 (34 ± 2) | 13–29 | 1.1–3.8 |

Total aboveground and belowground (down to 50 cm depth) C losses after rainforest conversion to plantations. Net biomass C uptake and time-averaged C stocks in biomass assuming linear increase of biomass over time. Carbon efficiency as unit of C lost (Mg C ha$^{-1}$) per unit of yield produced (Mg C ha$^{-1}$ y$^{-1}$). NPP tradeoff as unit of yield produced (Mg C ha$^{-1}$ y$^{-1}$) per unit of NPP lost for ecosystem functioning (NPP$_{eco}$)
[a]Mean ± SE ($n = 8$)
[b]Range of replicates
[c]Total C losses measured (total C losses considering time-averaged biomass)
[d]Total biomass without yield (only aboveground biomass)
[e]Total biomass time-averaged C stocks (only aboveground biomass)

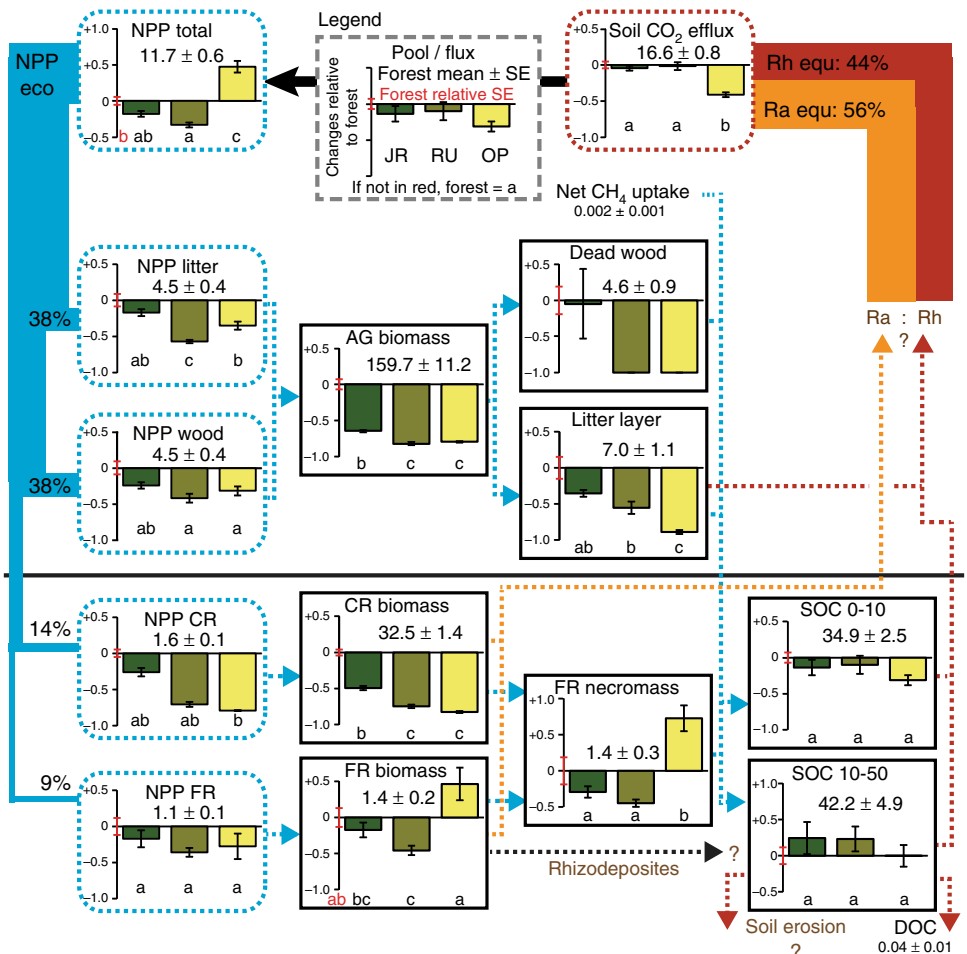

**Fig. 2** Effects of land-use change on ecosystem carbon pools and fluxes. Relative changes in plantations were calculated as (plantation value/forest mean)−1. Zero indicates no change compared to rainforest and −1, a decrease of 100%. Legends of individual figures are explained in the top central figure, where JR is jungle rubber, RU is rubber monoculture, and OP is oil palm plantation. Abbreviations correspond to net primary production (NPP), coarse roots (CR), fine roots (FR), aboveground (AG), soil organic carbon (SOC), dissolved organic carbon (DOC), contribution of autotrophic respiration (Ra equ), and heterotrophic respiration (Rh equ) in rainforest assuming the ecosystem to be at equilibrium. Dashed arrows indicate fluxes of C within the ecosystems. Fluxes in brown followed by a question mark represent the main knowledge gaps related to the C cycle. Means ± SE of forest pools and fluxes are given in Mg C ha$^{-1}$ or Mg C ha$^{-1}$ y$^{-1}$. Significant differences are based on ANOVA ($n = 8$) performed on absolute values (Supplementary Table 1)

corresponds to NPP$_{litter}$ plus NPP$_{roots}$ (assuming that root biomass pools have reached equilibrium in mature plantations), decreasing by at least 73 ± 5% and 65 ± 6% in rubber and oil palm plantations, respectively, as compared to rainforest (Fig. 3). Management of dead oil palm fronds led to a heterogeneous spatial distribution of aboveground C input, which is limited to the small fraction of the surface area occupied by frond piles (about 15%). The majority of the plantation area receives only belowground C input from roots, experiencing a dramatic decrease of 90 ± 5% of fresh C available for heterotrophs as compared to rainforest. The maximum biomass returning to the soil in zones outside the frond piles represented only 7 ± 1% of the total biomass produced in the plantation. The high productivity of oil palms associated with the removal of a large proportion of their production out of the plantations exacerbated the imbalanced share of NPP between human and ecosystem benefits compared to rubber plantations.

CO$_2$ efflux from soil under jungle rubber and rubber monocultures was similar to that in rainforest, despite having lower C input to the soil (Fig. 2)[30]. In contrast, soil CO$_2$ efflux was much lower (by 41 ± 3%) in oil palm plantations as compared to other land-use types. Unlike other land-use types, soil CO$_2$ efflux in oil palm plantations did not include CO$_2$ efflux from

litter decomposition because no chambers were located within frond piles. CO$_2$ efflux from litter decomposition can be approximated by litter production since C stocks in frond piles of mature plantations do not vary with age[27] and SOC accumulation under frond piles is low[23]. Soil CO$_2$ efflux plus litter CO$_2$ efflux (=soil CO$_2$ efflux + NPP$_{litter}$) in oil palm plantations would still be 23 ± 5% lower than in rainforest. Net methane uptake by methanotrophic bacteria in rainforest soil (1.9 ± 1.1 kg C ha$^{-1}$ y$^{-1}$) was three to four orders of magnitude lower than soil CO$_2$ efflux[30]. Thus, impacts of rainforest conversion on CH$_4$ uptake were negligible in terms of C fluxes and greenhouse gas emissions.

**Carbon pool dynamics.** While C stocks presented above are a snapshot in mature monocultures, the size and intensity of fluxes among C pools vary over the plantation lifetime. Losses of biomass are the highest following land clearing to establish monocultures and slightly decrease during plantation maturation due to C accumulation in the biomass of growing trees, which highly depends on plantation rotation time. To account for biomass pools dynamics in monocultures, time-averaged C stocks over a typical plantation lifetime were estimated assuming a linear

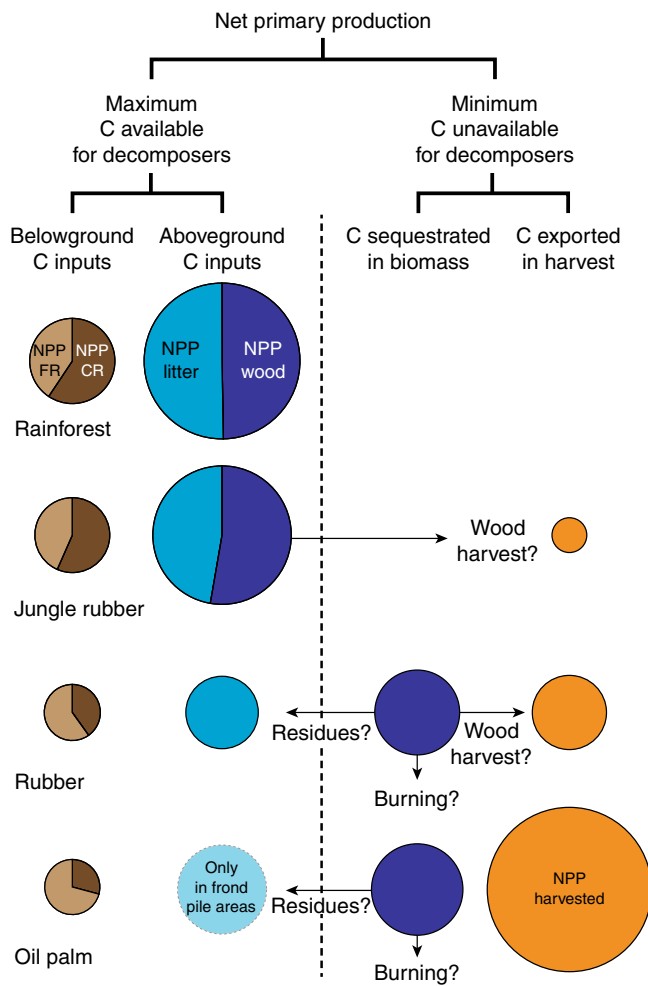

**Fig. 3** Share of net primary production between biomass pools available or unavailable for decomposer food webs in the land-use types. Biomass production of fine roots (NPP FR), coarse roots (NPP CR), litter (NPP litter) and wood (NPP wood) will be eventually fully decomposed in the ecosystem in opposite to the harvested (NPP harvested) and wood (NPP wood) biomass in monocultures. Arrows represent various fate of wood biomass at the plantation renovation depending on farmers' practice

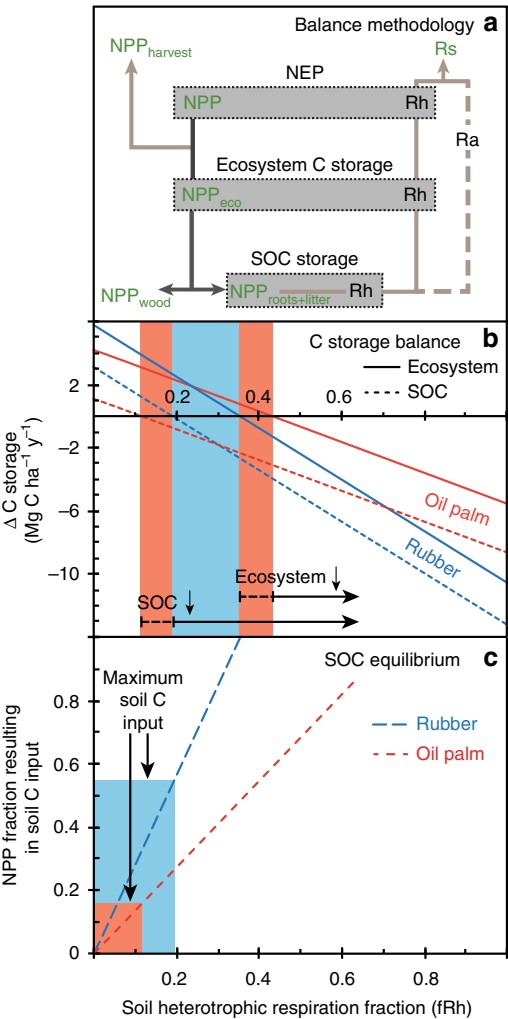

**Fig. 4** Changes in ecosystem and soil organic carbon storage depending on soil heterotrophic respiration. **a** Summary of the fluxes that must be balanced to estimate net ecosystem productivity (NEP), ecosystem C storage, and soil organic C storage (SOC). Fluxes in green were measured, i.e., litter, fine, and coarse root production ($NPP_{roots+litter}$), wood production ($NPP_{wood}$), the sum of the previous fluxes ($NPP_{eco}$), the harvest ($NPP_{harvest}$), the total net primary production (NPP) and soil $CO_2$ efflux (Rs). Balances depend on the contribution of heterotrophic (Rh) and autotrophic (Ra) respiration in Rs. **b** Ecosystem (solid lines) and soil organic carbon (SOC, dashed lines) balances in rubber and oil palm plantations depending on the fraction of soil heterotrophic respiration (fRh) in soil $CO_2$ efflux. Blue and orange areas correspond to fRh in, respectively, rubber and oil palm plantations at which ecosystem C storage would increase despite a decrease in SOC stocks. Soil organic C stocks balances were calculated for maximum soil C inputs. **c** Fraction of heterotrophic respiration at which SOC pool would be at equilibrium depending on actual soil C inputs (expressed as fraction of $NPP_{eco}$) in rubber (blue) and oil palm plantations (orange). Maximum soil C input corresponds to $NPP_{roots}$ in oil palm plantations outside frond piles and, additionally, $NPP_{litter}$ in rubber plantations if coarse and fine roots biomass pools in mature plantations are at equilibrium

increase in biomass during typical rotation times of rubber monocultures (40 years) and oil palm plantations (25 years); i.e., the biomass C stocks at the middle of one rotation[31]. Time-averaged biomass C stocks in oil palm plantations ($41 \pm 2\,\mathrm{Mg\,C\,ha^{-1}}$) were lower than in rubber plantations ($62 \pm 4\,\mathrm{Mg\,C\,ha^{-1}}$; Table 1) despite a similar net biomass C uptake in the vegetative parts of oil palms ($3.3 \pm 0.1\,\mathrm{Mg\,C\,ha^{-1}\,y^{-1}}$) and rubber trees ($3.1 \pm 0.2\,\mathrm{Mg\,C\,ha^{-1}\,y^{-1}}$, Table 1). Hence, over multiple rotations, converting rainforest to oil palm plantations leads to more ecosystem C storage losses ($-173 \pm 14\,\mathrm{Mg\,C\,ha^{-1}}$) as compared to converting rainforest to rubber plantations ($-134 \pm 18\,\mathrm{Mg\,C\,ha^{-1}}$; Table 1). Unlike in monocultures, biomass C pools dynamics in jungle rubber does not depend on plantation age but rather on the management of the native vegetation that was not cleared because rubber trees represented only 21% of the total biomass and 19% of the NPP.

On the long run, the C sink resulting from C accumulation in tree biomass of monocultures may be offset by the reduction in litter and SOC stocks. Due to the absence of time-series data and the lack of consistent data in the literature, SOC stocks instead of time-averaged SOC stocks were included in the estimation of ecosystem C storage losses. Ecosystem and soil C dynamics were estimated by balancing C inputs into the plant biomass or into

the soil with C outputs from soil $CO_2$ efflux (Fig. 4a). Mature oil palm plantations acted as C sinks because C outputs by soil + litter $CO_2$ efflux were $24 \pm 6\%$ lower than C input in the biomass ($NPP_{tot}$), even though soil $CO_2$ efflux included an autotrophic component, i.e., root and rhizomicrobial respiration. This results in positive NEP independently of the fraction of autotrophic $CO_2$

from soil (fRa). To determine if C accumulation in the biomass is offset by SOC losses, i.e., ecosystem C storage at equilibrium, the soil heterotrophic component of $CO_2$ efflux must be equal to the production of non-harvested biomass ($NPP_{eco}$). Ecosystem C storage in mature oil palm and rubber plantations increases only if heterotrophic $CO_2$ contributes less than $45 \pm 4\%$ and $36 \pm 3\%$, respectively to total soil $CO_2$ efflux (Fig. 4b, Supplementary Table 2). The fraction of heterotrophic $CO_2$ (fRh) in total soil $CO_2$ efflux was not determined. However, the fraction of $CO_2$ from heterotrophs reported in previous studies for rubber[32] and oil palm[33] monocultures ranged from 0.31 to 0.46 and from 0.20 to 0.70, respectively. Applying these portions to our sites, ecosystem C storage changes between $-1.7$ and $+0.7\,\mathrm{kg\,C\,ha^{-1}\,y^{-1}}$ in rubber plantations and $-2.6$ and $+2.3\,\mathrm{Mg\,C\,ha^{-1}\,y^{-1}}$ in oil palm plantations (Fig. 4b). It indicates that SOC losses occur and offset, at least partly, the C accumulation in the biomass.

Soil C stocks in rubber monocultures and oil palm plantations outside of frond piles would be at equilibrium if $CO_2$ from heterotrophs contributed to $19 \pm 1\%$ and $12 \pm 2\%$, respectively (Fig. 4b). These portions apply to a situation with maximal C inputs into the soil, i.e., the sum of coarse roots ($NPP_{coarseroots}$) plus fine root production ($NPP_{fineroots}$) in oil palm plantations, and additionally of litter production ($NPP_{litter}$) in rubber monocultures (Fig. 4c). SOC losses estimated with the above-mentioned fRh ranges reached from $-2.0$ to $-4.4\,\mathrm{Mg\,C\,ha^{-1}\,y^{-1}}$ in rubber plantations and $-0.8$ to $-5.7\,\mathrm{Mg\,C\,ha^{-1}\,y^{-1}}$ in oil palm plantations outsides of frond pile areas. Litter decomposition (measured by litterbags), SOC decomposition (measured as microbial respiration), and microbial activity (measured as microbial respiration per gram of microbial biomass) were lower in monocultures as compared to rainforest[34] (Supplementary Table 1). Despite the slowdown of dead organic matter decomposition and smaller soil $CO_2$ efflux in mature oil palm plantations, SOC stocks were still decreasing in mature monocultures.

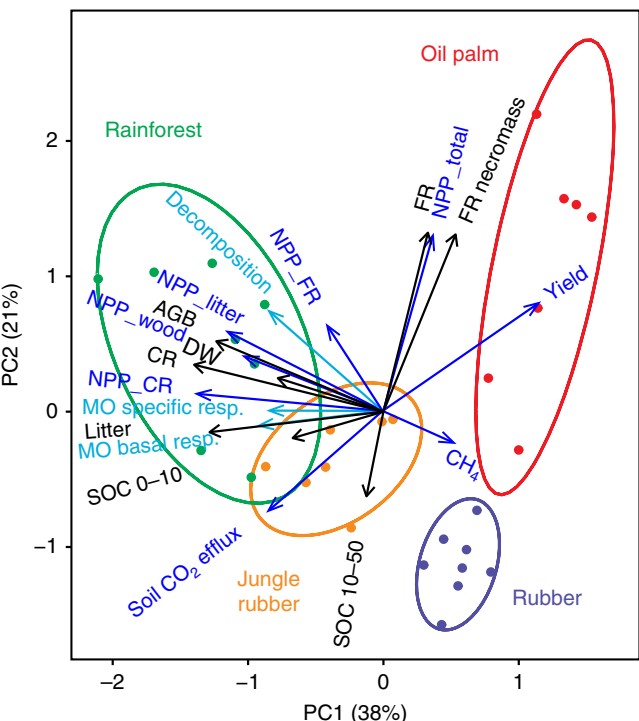

**Fig. 5** Ordination of research sites by 19 variables in principal components analysis. Stocks are in black, fluxes in dark blue, and decomposition processes in light blue. Abbreviations correspond to net primary production (NPP), aboveground biomass (AGB), coarse roots (CR), fine roots (FR), soil organic carbon between 0–10 cm depth (SOC 0–10) and 10–50 cm depth (SOC 10–50), methane uptake ($CH_4$), litterbag decomposition (decomposition, % of initial dry mass), microbial basal respiration (MO Basal resp.; $\mu g\, O_2\, g\, soil^{-1}\, h^{-1}$), and microbial-specific respiration (MO specific resp.; $\mu l\, O_2\, mg\, Cmic^{-1}\, h^{-1}$). Two research plots out of 32 were excluded because of incomplete datasets. Ellipses indicate 85% probability that an additional site falls within the ellipse of the respective land use

**Land-use tradeoffs.** Land-use type was the main factor explaining the total variability of C stocks, fluxes, and organic matter decomposition among sites (47%), while the effect of region (the design was duplicated in two regions of the province) was negligible (5%) according to a redundancy analysis constrained with land use and region (Supplementary Fig. 1). Each land use represented a possible pathway between the production of goods and maintenance of C stocks (shown by the first principal component (PC1), Fig. 5). Carbon stocks losses and the decrease in biomass production remaining in the ecosystem were limited in jungle rubber, but the yield was low as compared to monocultures. High yields in monocultures were associated with a strong decrease in C stocks and biomass production remaining in the plantation, and the slowdown of decomposition rates and soil microbial activity. Oil palm and rubber monocultures were distinguished by the second principal component (PC2), opposing biomass production and decomposition. It revealed a noteworthy relationship between $NPP_{total}$ and the quantity of fine roots biomass, contributing together with the yield to 49% of the variability explained by PC2 (Supplementary Fig. 2). The fine root necromass, litterbag decomposition, and soil $CO_2$ efflux contributed to an additional 32%. Oil palms maintained higher NPP and fine root density than any other ecosystem, despite a strong decrease in biomass C stocks and C allocation to the other vegetative parts. Oil palm plantations, however, induced the lowest C losses and reduction in biomass production remaining in the plantation per unit of harvested biomass (Table 1). Yield variation in jungle rubber and rubber monocultures was up to ten times higher in the most intensively harvested

plantations as compared to the least intensively harvested ones of each plantation type (Table 1). The efficiency of the most productive rubber monoculture was close to the least efficient oil palm plantations ($30\,\mathrm{Mg\,C\,loss\,ha^{-1}}$ vs. $29\,\mathrm{Mg\,C\,loss\,ha^{-1}}$ per $\mathrm{Mg\,C\,yield\,ha^{-1}\,y^{-1}}$).

## Discussion

The strong decrease in ecosystem C storage after rainforest conversion to perennial plantations resulted mainly from AGB loss. Nonetheless, overlooking C losses from belowground C pools, as it is currently the case with the IPCC Tier 1 method[24], results in a strong underestimation (up to 21%) of the C lost following land-use change. Furthermore, time-averaged biomass C stocks in oil palm ($41\,\mathrm{Mg\,C\,ha^{-1}}$) and rubber monocultures ($62\,\mathrm{Mg\,C\,ha^{-1}}$) were much lower than the default C stocks for oil palm ($68\,\mathrm{Mg\,C\,ha^{-1}}$) and rubber ($89\,\mathrm{Mg\,C\,ha^{-1}}$) monocultures used to estimate C losses in the IPCC Tier 1 method[24]. Our updated estimations were in line with C stocks reported in previous studies for time-averaged AGB of oil palm ($30$–$42\,\mathrm{Mg\,C\,ha^{-1}}$) with 25 years rotation time[27,35,36] and time-averaged biomass for rubber ($65\,\mathrm{Mg\,C\,ha^{-1}}$) with 38 years rotation time[37]. These stocks are also much lower than the threshold of $75\,\mathrm{Mg\,C\,ha^{-1}}$ initially defined by the HCS approach under which land converted into plantations would meet the C neutrality criteria[38] and still adopted by certification bodies such as RSPO[25] and in

scientific publications[39]. Calculation methodologies may substantially influence biomass estimations, depending on (1) allometric equations as it was the case for the biomass data in oil palms updated using recent and more robust equations[27,28] and (2) plantation rotation time that depends on regions and practices[40] because net C biomass uptake did not differ between rubber and oil palm plantations (Table 1). Despite uncertainties in plant biomass calculation, estimates from a growing body of recent literature remain far below default biomass stocks adopted for policymakers.

Calculating C stock change without accounting for SOC change in mineral soils implicitly assumes that monoculture ecosystem C storage increases after rainforest conversion because of the continuous C accumulation in biomass, i.e., plantations act as C sinks. Disregarding ecosystem C storage changes in oil palm and rubber monocultures because of the unknown soil heterotrophic $CO_2$ fraction, our C budgeting approach clearly indicates that SOC stocks are decreasing in mature plantations, offsetting at least partly the C accumulation in biomass. The approach included conservative estimates because it is based on maximum potential soil C inputs (Fig. 4c), i.e., assuming root biomass at equilibrium, which is unlikely for coarse root biomass in mature plantations[37,41,42]. Additionally, the range of heterotrophic contribution in soil $CO_2$ efflux under oil palms was reported for individual management zones and soil moisture[33], suggesting a more narrow range for soil $CO_2$ efflux averaged over a year at plantation scale. The fraction of soil heterotrophic $CO_2$ was extracted from a study on rubber monocultures with higher tree density and older age than the studied plots[32]. The reported range likely underestimates the contribution of heterotrophic $CO_2$ at our sites because it implies that roots respiration in rubber monocultures is similar to roots respiration in rainforest despite strong decrease in coarse and fine root biomass (Supplementary Table 2). Leaf C input may mitigate SOC losses in the 15% of the plantation area covered by frond piles. However, fronds are mostly mineralized aboveground and only a small fraction of their C contributes to SOC[23]. The main missing C flux in our study was the C lost by soil erosion in monoculture sites[18]. To our knowledge, no studies quantified SOC losses by erosion in oil palm and rubber monocultures. However, mature rubber and oil palm plantations in Malaysia lost on average 70 Mg of soil ha$^{-1}$ y$^{-1}$[43]. Applied to our sites that have on average 3.6% of C in the topsoil, it would result in 2.5 Mg C ha$^{-1}$ y$^{-1}$ exported from mature plantations by soil erosion. Other missing components were negligible in terms of C budgeting. Dissolved organic C (DOC) export in all land-use types was one order of magnitude lower than the standard error on soil $CO_2$ efflux[44]. Rhizodeposition, estimated to be 50% of fine root production[45], would be similar to the standard error of total NPP, and most of the C input is directly respired by rhizosphere microorganisms.

The lack of consistency in findings related to SOC dynamics might be due to limitations inherent to the space-for-time substitution approach used in most studies. Determining small SOC changes relative to large stocks is highly uncertain because spatial variability of SOC is high[21] and many factors such as clay content[17], initial land-cover[22] and C stocks[17], management practices[46], or business model[22] must be controlled. The uncertainty is exacerbated in studies that do not include natural sites as reference or immature plantations when investigating SOC dynamics in plantations chronosequence[22,47] because SOC stocks follow an exponential decay with most losses occurring during the first years after land-use change[17,48].

Even though SOC losses in the first generation of intensive plantations of relatively young age (maximum 17 years old) were small relative to the large SOC stocks stored down to 50 cm depth and more difficult to detect than biomass C losses, they were offsetting C accumulation in tree biomass and should be accounted for. Previous study in the same region reported significant SOC losses (14 Mg C ha$^{-1}$) under oil palms[17], similar to the non-significant decrease found in our study (11 Mg C ha$^{-1}$), and representing one-third to one-half of the C accumulated in tree biomass. Our approach shows that SOC equilibrium had not been reached yet. Previous research at our study sites showed that the fraction of labile SOC pools in total SOC was lower in plantations[49]. Despite a decrease in soil microbial activity, microorganisms were relying on hardly decomposable SOC pools with a long turnover to mitigate the strong decrease in fresh C inputs and maintain microbial metabolism. Since soil across the majority of oil palm plantation area received only 10% of the C inputs of those in rainforests, the equilibrium of SOC stocks will be at a very low level. The large imbalance between soil C inputs and outputs, and the fact that soil erosion does not stop in mature plantations[43] indicate an increase in the relative contribution of SOC losses to total C losses over time and over multiple rotation cycles.

Oil palm cultivation had the most negative impact on ecosystem C storage mostly because of the shorter rotation time of oil palm plantations as compared to rubber monocultures. The positive NEP of oil palm plantations resulted mostly from the high biomass production harvested, yet the C fixed in this biomass pool is exported from the plantation. Since C is sequestered only for a short time in food, biodiesel, or oleo-chemical products, harvested biomass production is considered as C neutral[50]. Oil palm was the most efficient land-use type in the tradeoff between regulating and provisioning ecosystem services related to C. Oil palm cultivation led to the smallest C stock losses and smallest decrease in the produced biomass remaining in the ecosystem per unit of harvested biomass (Table 1). About 96% of the C losses due to forest biomass clearance in Indonesia is emitted in the atmosphere in the first 15 years because forest clearance yields a high proportion of non-merchantable wood or wood used for energy and paper production[51]. Favoring wood extraction for long-term use prior to land clearing would strongly reduce GHG emissions from rainforest conversion. Extensive cultivation forms, such as jungle rubber that allows growth of native vegetation within plantations, also limit C losses (Fig. 1). However, substantial changes in stand structure[29], especially the absence of tall trees, still lead to large C losses. Despite low yield in extensive rubber plantations, the ranges of C efficiency and NPP tradeoff in extensive and intensive rubber plantations overlapped. Wood extraction in jungle rubber and the use of rubber tree wood at the end of the plantation rotation were not monitored and included in the harvested biomass, which would almost double the yield of rubber monocultures. When rubber monocultures are not burned at the end of each cycle, the range of C efficiency of rubber cultivation would overlap with the one of oil palm cultivation.

The human appropriation of NPP decreases the C available for ecosystem functioning and decomposer food webs[52], amplified by the absence of wood turnover (Fig. 3). Besides a reduction in soil microbial biomass and activity, and litter turnover, previous research at our study sites have shown shifts in trophic links and reduced energy flow through decomposer food webs[53,54]. Consequently, impacts resulting from reduced C inputs due to land-use intensification cascade to higher trophic levels and key ecosystem functions such as C and nutrient recycling. Nonetheless, oil palm cultivation offers room to re-equilibrate the balance between human and ecosystem needs. Only a quarter of the harvested biomass is transformed into palm oil, leaving a substantial amount of biomass available for other uses such as energy source or organic fertilizers[55]. Long-term application of empty fruit bunches (EFB) has positive impacts on soil fauna, soil

fertility, and oil palm yield, underlying the importance of maximizing organic matter inputs to soils[46,56]. Since oil palm trunks cannot be used for timber and have little economic value[55], they represent a potential source of organic matter input. Accordingly, oil palm C efficiency can be improved to mitigate conflicts between economic and environmental needs.

Eco-efficiency of a land-use type, however, cannot be evaluated solely on the basis of C resources. Oil palm productivity was supported by higher inputs and consumption of natural resources. Mineral fertilizers and herbicides were systematically applied to oil palm plantations but only occasionally to rubber plantations[14]. Previous research in these study sites shows that water consumption of oil palms was also higher than that of rubber trees, reaching the water consumption of rainforest[57]. Despite having lower biomass, oil palms tended to invest more in fine roots biomass than plants in rainforest (Fig. 2), suggesting that such high NPP cannot be sustained without elevated water and nutrient consumption by fine roots. High resource consumption increased agricultural yield, but negative impacts have already been observed in the landscape, e.g., water scarcity during the dry season[57], decrease in water quality, and high nutrient leaching from oil palm plantations[14].

In conclusion, a comprehensive dataset was essential to highlight the full range of C gains and costs of tropical land-use intensification because of multiple and diverse pools, fluxes, and impacts on the C cycle following conversion of rainforest to plantations. The short-term C sequestration potential in the plantation biomass was lower than default values currently used to assess their sustainability or their impact on GHG emissions and is partly or even totally offset by soil C losses. Despite small relative effects of land-use change on SOC compared to the large effects on biomass in the short-term, soil C should not be overlooked because its slow dynamics imply delayed impacts. When the high C stock losses that follow plantation establishment are put into perspective with the high productivity, oil palm cultivation is the most efficient land-use type in terms of C resources due to high fertilizer and herbicide application, disregarding other resources. This stresses the importance of evaluating tradeoffs between ecosystem functions. Thorough assessments of land-use impacts on resources such as biodiversity, nutrients, and water must complement this synthesis on C but are still not available. Indeed, high consumption of natural or external resources by intensive plantations has impacts on ecosystem functioning that cast doubt on their sustainability. Land-use intensification should not reach the threshold where increasing external inputs become necessary to compensate for decreasing resource recycling by internal ecosystem processes[58], thereby entering a feedback loop of intensification to maintain current land-use productivity, rather than to increase it.

## Methods

**Study sites.** The studies were conducted in the lowlands of Jambi province in central Sumatra, Indonesia, within the interdisciplinary EFForTS project[59]. The climate is humid tropical (27 °C, 2235 mm y$^{-1}$) with a drier period in July–August. Four prevalent land-use types in this area, namely (1) rainforest, (2) jungle rubber, (3) rubber monocultures, and (4) oil palm monocultures, were investigated in two landscapes having contrasting soil texture; Harapan with loamy Acrisols and Bukit Duabelas with clayey Acrisols. The lowland rainforest sites were close to natural state but were slightly affected by selective logging and extraction of non-timber rainforest products in the past. Jungle rubber plantations were smallholder rubber agroforests, which were established by planting rubber trees (*H. brasiliensis*) into partly logged rainforests. Rubber plantations (7–17 years old) and oil palm (*E. guineensis*) plantations (9–16 years old) were smallholder monocultures. Farmers harvest latex by tapping rubber trees several times per week or oil palm fruit bunches 1–2 times a month all year around. During harvest, oil palm fronds are stacked in piles every second avenue. Oil palm plantations were fertilized once or twice per year with 300–550 kg NPK-fertilizer ha$^{-1}$ y$^{-1}$[60]. Potassium chloride, urea or dolomite may be applied occasionally. However, rubber monoculture had not been fertilized the year measurements took place, as it is commonly the case for

smallholder rubber plantations. Chemical and manual weeding took place all year round in rubber and oil palm monocultures. In each landscape, four 50 × 50 m$^2$ replicate plots for each land-use type were selected, for a total of 32 plots. The experimental design consisted of a space-for-time substitution approach to assess the effect of the land-use change from rainforest to plantations. Sites were carefully selected based on climatic conditions, vegetation (for rainforest sites), soil conditions, and being located on similar landscape positions so that any differences could be attributed solely to land-use effects.

**Biomass and necromass carbon stocks.** Stand structural parameters were recorded on all 50 × 50 m$^2$ plots for each tree, palm, and liana with a diameter at breast height (DBH) ≥10 cm. Understory trees with a DBH of 2–9.9 cm were inventoried in the same way on two 5 × 5 m$^2$ subplots. Diameter was measured at 1.3 m and tree height was recorded using a Vertex III height meter (Haglöf, Långsele, Sweden). Wood density values were either determined directly from wood cores extracted from 204 trees or interpolated values were applied using a calibration equation based on pin penetration depth measured with a Pilodyn 6J wood tester were applied (PROCEQ SA, Zürich, Switzerland). Allometric equations were used to estimate AGB and coarse roots biomass for forest trees[28,61], rubber trees[62], and oil palms[27,63]. Refer to Kotowska et al.[29] for all methodological details. Fine roots biomass (diameter: ≤2 mm) was measured using 10 soil cores (3.5 cm in diameter, 50 cm soil depth) in each plot, which were located in a randomly placed grid. In oil palm plantations a higher proportion of roots necromass can be expected due to management such as glyphosate application, pronounced seasonality in fine roots mortality[64], and slower decomposition rates. All fine roots segments >1 cm lengths were extracted by washing on a sieve and separated under a stereomicroscope into live (biomass) and dead (necromass) fractions. Alteration in periderm color, non-turgid cortex, root elasticity, and the absence of living root tips were used as determinants for root death. Woody coarse debris were analyzed within all forest and jungle rubber plots, where snags (DBH > 10 cm) and logs (mid-point diameter: >10 cm, length: >1 m) were recorded.

**Net primary production.** Aboveground litterfall, pruned oil palm fronds, rubber latex harvest, oil palm fruit harvest, and stem-increment were measured from March 2013 to April 2014. Litter from 16 traps per plot was collected at monthly intervals and sorted into leaves, woody material, propagules, and inflorescences, which were subsequently oven-dried for 72 h at 60 °C. In the oil palm plantations, all pruned palm fronds were counted. The yield of oil palm fruits and rubber latex (in Mg ha$^{-1}$) was recorded by weighing the harvested material for all trees in each plot. Woody biomass production was calculated from pairwise difference in tree biomass between measurement dates based on the above-mentioned allometric regression models. For trees, stem-increment data were based on manual dendrometer tapes (UMS, Munich) placed on 40 trees per plot (960 trees in total). The suitable trees were chosen randomly from three size groups (small, medium, large) accounting for the system-specific size distribution and allowing for a higher proportion of large trees (>40 cm)—if present—as they contribute a major share of total biomass. Oil palm biomass production was derived from stem height growth measured for each palm. Fine roots production was measured using an ingrowth core approach with 16 cores per plot. The extracted soil cores were processed in the same manner as for the fine roots inventory. To convert biomass into carbon units, the C content of stem wood, fine roots, dead wood, rubber latex, oil palm fruit, and all litter fractions were analyzed with a CN analyzer (Vario EL III, Hanau, Germany). For all methodological details, see Kotowska et al.[29].

**SOC stocks.** SOC stocks were determined between October and November 2012. Soils were described and sampled per horizon (horizons A, E, Bt1, Bt2, and horizons of transition if present) down to 50 cm depth in one soil pit for each of the 32 plots. Total carbon content was measured at the University of Göttingen with an elemental analyzer (Eurovector) coupled to an isotope ratio mass spectrometer (Delta plus, Thermo Fisher). Because of the absence of carbonate in the heavily-weathered soils, total C content was equal to organic C content. Bulk density was determined using 250 cm$^3$ cylinders inserted at 5, 20, and 40 cm depths, and C stocks were calculated by multiplying C content by respective bulk density over the thickness of the soil horizon. For more details, see Guillaume et al.[18].

**Litter decomposition and microbial activity.** Litter decomposition was measured using litterbags. Mass loss was calculated as the difference between initial litter dry mass and litter dry mass remaining after 12 months. Litterbags (20 × 20 cm$^2$ with 4 mm mesh size) containing 10 g dry leaf litter were incubated in the field from October 2013 to October 2014[65]. Leaf litter composition in the litterbags reflected fallen litter at the plot of exposure. For rainforests, a mixture of freshly fallen, senesced leaf litter of three tree species (40% *Garcinia* sp., 30% *Gironniera nervosa*, 30% *Santiria lavigata*) collected from one of the rainforest plots was used. For jungle rubber and rubber plantations, litter comprised freshly fallen, senesced leaf litter of the rubber tree (*H. brasiliensis*). For oil palm plantations, freshly cut leaves (ca. 15 cm, without leafstalk) from oil palm (*E. guineensis*) were selected. For more details, see Krashevska et al.[65].

For measuring microbial respiration, soil (0–5 cm depth) was sampled with a 5 cm diameter corer[34]. Soil basal respiration and microbial biomass (substrate-

induced respiration[66]) were determined by measuring $O_2$ consumption using an automated respirometer system[67], for details, see Krashevska et al.[34]. Microbial-specific respiration was calculated using the basal respiration and microbial biomass data. Litter stocks per hectare were calculated from data on C content and amount of litter from Krashevska et al.[34], for details, see Drescher et al.[59].

**Soil $CO_2$ and $CH_4$ fluxes.** Soil $CO_2$ and $CH_4$ flux data were taken from Hassler et al.[30], who measured fluxes from December 2012 to December 2013 in the same research sites. In oil palm plantations, by chance, no chamber was located in a frond pile area because frond piles cover a relatively small area. Consequently, soil $CO_2$ efflux in oil palm plantations included autotrophic $CO_2$ from roots and rhizosphere and heterotrophic $CO_2$ from SOC mineralization but not from litter mineralization.

**Net ecosystem productivity and SOC stocks balance.** NEP and the dynamics of ecosystem and soil organic C storage at plantation scale were estimated depending on the fractions of soil heterotrophic $CO_2$ (Rh; $CO_2$ from SOC and litter decomposition), and soil autotrophic $CO_2$ (Ra; $CO_2$ from respiration of roots and rhizosphere microorganisms[68]):

$$NEP = NPP - Rh \qquad (1)$$

$$Rh = Rs * fRh \qquad (2)$$

where NPP is the net primary production ($Mg\,C\,ha^{-1}\,y^{-1}$), fRh is the fraction of heterotrophic $CO_2$ in soil $CO_2$ efflux (Rs; $Mg\,C\,ha^{-1}\,y^{-1}$). Positive NEP indicates C sinks. To assess equilibrium of ecosystem C storage, $NPP_{eco}$, i.e., NPP minus harvested biomass, replaces NPP. Combining Eqs. (1) and (2) we obtain:

$$NEP = NPP - Rs * fRh \qquad (3)$$

The fraction of heterotrophic $CO_2$ was not determined in the field. Therefore, ecosystem C storage in monocultures was expressed in function of the fraction of heterotrophic $CO_2$ in soil $CO_2$ efflux using Eq. (3) and compared with fRh from literature. Heterotrophic $CO_2$ effluxes from dead wood in rainforest and jungle rubber were not measured. Assuming all C pools to be at equilibrium in rainforest ecosystem (NEP = 0), $NPP_{wood}$ was subtracted from NPP to estimate fRh in rainforest. Since this assumption is not applicable for jungle rubber, this methodology could not be applied to this land-use type. Finally, in oil palm plantations, Rs did not include $CO_2$ efflux from litter decomposition in frond piles. Therefore, $NPP_{litter}$ was added to Rs to include $CO_2$ losses from frond piles mineralization, assuming biomass C input and $CO_2$ output from the decomposition of fronds at equilibrium over 1 year.

Soil organic C stocks change ($\Delta$SOC) depend on the balance between soil C inputs and soil C outputs:

$$\Delta SOC = C\ inputs - C\ outputs \qquad (4)$$

Soil C inputs result from dead biomass (DB) inputs. Soil C outputs result from dead organic matter mineralization by soil heterotrophic organisms emitting $CO_2$ and DOC export in systems without soil erosion. Since biomass does not decrease in healthy monoculture, soil C input over a year cannot be higher than the NPP minus the harvested biomass ($NPP_{eco}$) over a year. Hence, the actual soil carbon input will be a fraction of $NPP_{eco}$. This fraction depends on the turnover of biomass pools. A fraction of biomass production results in a net increase of the biomass pool (fNPPi). The rest of the production compensates biomass mortality during its turnover (fcNPPt).

$$fNPPi + fNPPt = 1 \qquad (5)$$

$$NPPt = NPP_{eco} * fNPPt = NPP_{eco} * (1 - fNPPi) \qquad (6)$$

The pool NPPt corresponds to the amount of C potentially available for decomposer food chain and depends on the turnover of each biomass pool.

$$\begin{aligned} NPPt = NPP_{wood} * fNPPt_{wood} + NPP_{litter} * fNPPt_{litter} \\ + NPP_{coarseroots} * fNPPt_{coarseroots} + NPP_{fineroots} * fNPPt_{fineroots} \\ + NPP_{rhizodeposits} * fNPP_{rhizodeposits} \end{aligned} \qquad (7)$$

where $NPP_{litter}$ is the amount of leaves, fruits, twigs, etc. measured in litter traps and $fNPPt_{rhizodeposits}$ is the fraction of rhizodeposition not directly assimilated by soil microorganisms closely associated to roots or invested in microbial biomass growth. When a biomass pool is at equilibrium, i.e., the NPP of the pool compensates biomass mortality of the pool, fNPPt = 1. When a pool has no turnover of its biomass, i.e., the NPP of the pool results in net biomass increase, fNPPt = 0. Changes in dead biomass (DB) stocks are determined as:

$$\begin{aligned} \Delta DB = \Delta SOC + \Delta dead\ roots + \Delta litter + \Delta dead\ wood \\ = NPPt - Rh - DOC \end{aligned} \qquad (8)$$

In complex ecosystems such as rainforest or jungle rubber, it is difficult to determine changes in DB stocks (Rh) because of the difficulty to measure wood mortality ($fNPPt_{wood}$) and $CO_2$ efflux from dead wood mineralization by heterotrophic organisms. In monocultures, these difficulties are absent because there are no dying trees, i.e., no wood turnover ($fNPPt_{wood} = 0$). When measuring soil $CO_2$ efflux from a collar where the litter layer was not removed, the heterotrophic $CO_2$ component (Rh) of the efflux is a measure of the C output from the DB pool composed of SOC, litter, and dead roots. SOC output from DOC export are neglected because they are two to three orders of magnitude lower than soil $CO_2$ emissions (Fig. 2). Thus, in monocultures

$$\Delta SOC + \Delta litter + \Delta dead\ roots = NPPt - Rh \qquad (9)$$

Accordingly, all DB in monocultures are C inputs to these three C pools. Because $NPP_{litter}$ is measured by collecting litterfall, $fNPPt_{litter} = 1$ by definition. $NPP_{rhizodeposits}$ is rarely measured, resulting in an underestimation of total NPP and soil C input. Nonetheless, the error is small because (i) $NPP_{rhizodeposits}$ was estimated to be about 50% lower than $NPP_{fineroots}$[45], which itself represents 9–13% of $NPP_{eco}$ and (ii) $fNPP_{rhizodeposits}$ is small because the fraction of rhizodeposition directly mineralized by microorganisms closely associated to roots is large. The $CO_2$ produced by these microorganisms is accounted as autotrophic $CO_2$ and so the respired C is not a C input to the soil. Soil C inputs in monocultures of perennial crops can be simplified as follows:

$$\begin{aligned} C\ input = NPPt = NPP_{litter} + NPP_{coarseroots} * fNPPt_{coarseroots} \\ + NPP_{fineroots} * fNPPt_{fineroots} \end{aligned} \qquad (10)$$

Formally, the difference between NPPt and Rh indicates a change in a C pool composed of SOC, litter layer, and dead roots.

$$\begin{aligned} \Delta SOC + \Delta litter + \Delta dead\ roots = NPP_{litter} + NPP_{coarseroots} * fNPPt_{coarseroots} \\ + NPP_{fineroots} * fNPPt_{fineroots} - Rh \end{aligned} \qquad (11)$$

Even though it is not possible to determine from Eq. (11) in which of the three pools C stock changes occur, in practice stock changes can be attributed to SOC. Litter layer C stocks in mature rubber plantations are small and are in quasi-equilibrium, i.e., $\Delta litter \approx 0$. There is no clear limit when dead roots are considered as SOC. In practice, dead roots passing through a 2 mm sieve are included in the free particulate organic carbon fraction and considered as SOC. Even though $\Delta$dead roots is expected to be small for the same reason than $\Delta$litter, in this approach dead roots are considered as a SOC fraction, just as free particulate organic C or mineral associated C are SOC fractions. At SOC equilibrium

$$\Delta SOC = 0 = NPPt - Rh = NPP_{eco} * fNPPt - Rs * fRh \qquad (12)$$

with fRh the fraction of heterotrophic $CO_2$ in soil $CO_2$ efflux (Rs). SOC stock dynamics depend on the fraction of NPP resulting in soil C input and the fraction of heterotrophic $CO_2$ in soil $CO_2$ efflux. In rubber plantation, the maximum soil C input ($NPP_{max}$) occurs when root biomass pools are at steady-state, i.e., root biomass production results in equivalent soil C inputs ($fNPPt_{coarseroots} = fNPPt_{fineroots} = 1$):

$$NPPt_{max} = C\ input_{max} = NPP_{litter} + NPP_{coarseroots} + NPP_{fineroots} \qquad (13)$$

In oil palm plantations, there is a negligible amount of litter except in the frond pile area. When Rs is measured outside frond pile areas, the only C input to these areas comes from roots because $NPP_{litter} = 0$

$$NPPt_{max} = C\ input_{max} = NPP_{coarseroots} + NPP_{fineroots} \qquad (14)$$

The maximal fraction of heterotrophic $CO_2$ (fRh) in soil $CO_2$ efflux (Rs) at which SOC pool still can be at equilibrium ($\Delta SOC = 0$) is calculated using the maximal fraction of NPP resulting in soil C input (fNPPt):

$$fRh_{max} = \frac{NPP_{eco} * fNPPt}{Rs} \qquad (15)$$

where $fNPPt = NPPt_{max}/NPP_{eco}$. Any fRh above this limit results in SOC losses. Eq. (15) in oil palm plantations is limited to the analysis of SOC dynamics in the area outside frond piles. The comparison of fRh with literature values determines the plausibility of the SOC equilibrium. The actual fRh at which SOC would be at equilibrium is lower and depends on the pool turnover of coarse roots ($fNPPt_{coarseroots}$) and fine roots ($fNPPt_{fineroots}$), as well as on the fraction of $NPP_{litter}$ that is not mineralized over a year. Root-to-shoot ratio of mature oil palm and rubber tree stay relatively constant, implying an increase of root biomass[42,69,70]. However, fine root biomass of oil palms stays relatively constant[69]. This suggests high turnover of fine roots but low turnover of coarse roots, i.e., $fNPPt_{coarseroots}$ close to 0 and $fNPPt_{fineroots}$ close to 1.

**Statistics and calculations.** Statistical analyses were performed using R 3.2.3 software (R Core Team 2017). Because the landscape effect was in general not significant and explained only 5% of the full dataset variation, it was excluded from

the model and effects of land use on response variables were tested by one-way ANOVA using eight replicate sites per land-use type. Differences between group means were assessed by Tukey's HSD test. Normality and homoscedasticity of model residuals were tested using Shapiro–Wilk and Bartlett tests. If significant, data were log-transformed. When assumptions were not met after transformation, land-use effects were assessed with non-parametric tests (Kruskal–Wallis Rank Sum Test for more than two land-uses and Wilcoxon Test in the case of dead wood, which is present only in two land uses). Differences between land uses were tested by Pairwise Test for Multiple Comparisons of Mean Rank Sums (function posthoc.kruskal.nemenyi.test of the PMCMR package). Principal components analysis (PCA) was conducted on standardized data using the function prcomp and redundancy analysis (RDA) using the function rda. Data are presented as the mean of eight replicates ± standard error (Supplementary Table 1). The standard error associated with mean differences between land uses (e.g., total C losses) were calculated as the square root of the sum of the squared standard error associated with each of the compared land uses. If not specified, all discussed differences are significant at a $p$-value < 0.05.

**Data availability**. Data that support the findings of this study are archived at EFForTS-IS, with openly accessible, keyword-searchable metadata, and data holder contact details for data requests. Datasets used in this study have the identification numbers 11985 (soil carbon), 12002 (B04_Biomass and productivity), 12322 (decomposition), 12013 (basal respiration and microbial biomass)[71].

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

## Acknowledgments

This study was financed by the German Research Foundation (DFG) in the framework of the collaborative German–Indonesian research project CRC990. We thank the following persons and organizations for granting us access to and use of their properties: village leaders, local plot owners, PT Reki, and Bukit Duabelas National Park. We thank Evelyn Hassler for sharing the raw $CH_4$ and $CO_2$ data for each plot.

## Author contributions

T.G. conducted data analyses and lead the writing of the paper. Y.K. supervised the work. T.G. and M.K. assembled the dataset and produced figures. T.G., M.K., D.H., V.K., S.S., and Y.K. contributed data. T.G., M.K., D.H., A.K., V.K., K.M., S.S., and Y.K. contributed to interpretation and writing.

## Additional information

**Competing interests:** The authors declare no competing interests.

