## [Peer Review File · Nature Communications]

Reviewers' comments:

Reviewer #1 (Remarks to the Author):

This manuscript presents a carbon budget analysis for three tree plantation types in Indonesia (jungle rubber, rubber monoculture, and oil palm monoculture) and compares them with the carbon budget of natural forest. The topic is timely and the question being asked here is of great societal benefit. The authors present a very thorough C budget analysis. While I see the value of this work, the manuscript needs substantial revision. I summarize my key recommendations below:

1. The title and beginning of the abstract/introductions made me think this was a global study, when in fact it is focused specifically on tree plantations in Indonesia. The title should reflect this, and the abstract and introduction could also state this up front.

2. The abstract is extremely vague. It needs more specifics. I give examples in my detailed comments below.

3. I recommend including more background on the plantations studied, how they are managed, what do you mean by more intense management? Why did you choose these sites to address this larger question? Discuss net primary productivity and other aspects to help the audience follow the logic of the paper. This would help prepare the audience for the very detailed C budget review in the rest of the paper.

4. As a whole, I had a very difficult time following the paper. The authors present a LOT of information, a conceptual diagram showing the C pools analyzed would help. The introduction could also provide a road map to help lead the audience through the details of all of the analyses.

5. I think that the authors need to be very careful with how they are using different terms. For example, NPP is a rate, but the authors discuss NPP as if it were a pool of Carbon. It is important to be precise with language and to ensure that key terms are defined. NPP isn't "removed" from an ecosystem, the biomass is removed from the system, and then productivity declines.

6. In the NEP section, the authors do not account for woody respiration. This should be acknowledged. This is why definition of terms is really important. Some of the measurements made

are indirect measures of NPP and NEP and the assumptions being made need to be clearly spelled out.

7. There are quite a few grammatical errors in the paper that should be addressed.

All in all, I think the paper has a lot of potential. It is obvious that a lot of work went into compiling the carbon budgets, and I think the results are impactful; however, the presentation of the ideas is unclear, much of the discussion is vague, and the use of terminology is imprecise. I had a very difficult time following the logic of the paper and critical information on how the plantations are managed, what the authors mean by "yield", etc. was missing or described after the fact. As a result, the paper does not live up to its full potential.

Detailed Comments

Abstract

The abstract in general is overly vague. I recommend adding in quantitative information whenever possible, for example how much forested land (e.g., in hectares) does Intensive Oil Palm allow to remain forested? How unbalanced is the sharing of NPP? Strongly perturbs the functioning (what functions?), what other services?

The focus of the paper appears to be global, so the discussion of research plots in Indonesia is confusing here.

Efficient to produce yield? Net primary production loss for ecosystem functioning? I think this sentence needs rephrasing so that it is clear. Yield of what?

L41 crop makes me think of agricultural plants, like corn, but this paper is about conversion to tree plantations. Most crops have quick turnaround time, but tree plantations take several years to develop and the disturbances and environmental consequences associated with tree plantations are likely to be very different than that of traditional agricultural crops. I would keep the focus on what was studied.

Introduction

In reading the abstract and the beginning of the introduction, I thought this was a global analysis, but it focuses specifically on tree plantations in Indonesia. This needs to be more clear in the title and stated up front.

As a whole, more introduction into the plantation types studied is needed. Why were these land use types chosen? How are they managed?

I also wanted more introduction into the specific measurements being made. It is overwhelming to get to the methods section without a sense of all of the considerations that went into measuring the C pools and fluxes. What is it that makes this study unique?

L52 "increased" greenhouse gas emissions

L64 This is confusing. NPP isn't "available" for ecosystem function, it is an ecosystem function.

L76 a citation is needed here

L74 I disagree with this statement. There are research groups that perform whole system C budgets. They are often scarce because some numbers are difficult to get e.g., belowground pools can be difficult to quantify.

L75 spell out IPCC the first time

L76 can you quantify how much data are default versus real values? And are the defaults used from tropical systems?

L77 please spell out the acronym used here

L80 what is "plantation scale" do you mean by studying individual plantations?

L87 "reduced NPP remaining in the system" this needs to be rephrased. NPP is not a pool, it is a rate (e.g., Mg C/ha/year). Litterfall is a product of NPP (e.g., C fixed from the atmosphere). We use it as an indirect measure of above ground productivity. If litterfall is removed from a system, biomass is being removed, not NPP.

Results

The use of NPP is not quite right. This sounds to me like the authors are discussing carbon stocks and not fluxes.

L178 yield of what? What is being harvested exactly?

Discussion

L212 up “to” 40

L213 “on” average

L215 “On” the other hand

Methods

The tree species should be listed here rather than using the common names.

L343 how many jungle rubber plantations were studied?

L418 the authors are not accounting for woody respiration, which should be acknowledged and justified.

Reviewer #2 (Remarks to the Author):

This manuscript uses an extensive dataset on carbon stocks and fluxes across four land uses with high tree cover in Sumatra, Indonesia to evaluate how land use affects carbon storage dynamics in the ecosystem, as well as harvested carbon yields. The authors find that while oil palm has the most net carbon loss (compared to natural forest), it also yields the most carbon for human use. The work is of interest to academics and practitioners working in tropical land use change, and may inform design of land-based carbon projects (e.g., REDD+) [although the authors do not engage with this policy angle in their paper]. However, I had a hard time understanding the big novel finding from the work that would move the field forward, especially given that this group has previously published some of these data, including in Nature Communications (doi:10.1038/ncomms13137). Perhaps the most novel finding, in my opinion, is that soil organic carbon in these mineral soils appears to still be decreasing in oil palm plantations. The statistical analysis seems robust, and the authors provide detailed (for the most part) methods documenting their procedures that should allow for reproduction of these results if required. Beyond the issue of novelty, there are some limitations of the manuscript as currently written. I outline these general issues and specific comments below.

GENERAL COMMENTS

1. Clarity. In reading the manuscript, I often found that I was confused. The biggest improvement that the authors can make to the article, in my opinion, is in clarity of writing. One source of confusion was the difference between data that was collected/analyzed for this article, versus

previous findings by the group, versus findings of others. Placing these in context (e.g., “previous research in these study sites reported”... or “in Kalimantan, XX author found that”) would help immensely. In addition, only after I read the entire manuscript did it become clear that the authors did not measure heterotrophic respiration, but infer the heterotrophic/autotrophic partition. This is fine, but stating this clearly in the main text of the article is needed so that much of the results and discussion make sense!

2. Novelty. This manuscript brings together a huge amount of robustly-collected data to understand carbon fluxes in a region of Sumatra. However, the authors don't do a great job of highlighting what is novel about this work, especially compared to their previous publication in Nature Communications that also reports NPP, harvested biomass, and carbon stocks. In the introduction, the authors imply that there has not been region-specific accounting of carbon fluxes under land use change to oil palm and rubber. There are many studies that tackle this problem, but these other studies are mostly not acknowledged by the authors, nor are their shortcomings (and they have shortcomings!) discussed. The authors might want to grapple specifically with the contrasting findings from (van Noordwijk et al. 2015) and (van Straaten et al. 2015), or the C measurements of (Koh and Ghazoul 2010) – I'm sure there are others too. This would better highlight the novelty of the work, and its contribution not just to “advising tropical countries in their development policies”, but also to the theoretical and empirical understanding of carbon dynamics in tropical tree plantations. Personally, I think that the contribution to understanding soil C stock dynamics (including inputs and removals) as well as the partitioning among yield versus ecosystem NPP is fascinating, but the authors should determine what they think is novel about their manuscript, in the context of other literature.

3. Land Sparing. In the abstract, the authors claim that oil palm spares forested land. This statement is not supported by the manuscript, and is highly contextual (e.g., see recent review by C. Kremen in the Annals of the New York Academy of Sciences, as well as Carrasco et al. paper on increasing oil palm yields in Science). I would revise the abstract to avoid grappling with this sticky topic.

4. Justification for Statistical Approach. While I think the statistical approach is likely a good choice, I'd like to see a justification for these tools in the methods – just a few sentences will ground the methods and argue for why they are appropriate to the data at hand.

SPECIFIC COMMENTS

Abstract. In the abstract, it would be great to see more results from the paper, especially novel findings or specific values. As currently written, the abstract contains just one sentence describing findings from this research.

Line 56. “The tropics has a lower eco-efficiency”. It seems that the tropics *could* have high yields, and in some cases, they do (how about soy in Brazil?). Also, there are other ways to measure these tradeoffs beyond just carbon. See e.g. (Pittelkow et al. 2013), (Mueller et al. 2012), (Brauman et al. 2013), (Carlson et al. 2017) - perhaps the authors need to define eco-efficiency more specifically related to carbon.

Line 60. “phytosanitary products for oil palm cultivation” – this is the first time in the main text that oil palm has been introduced, so it feels uber-specific – consider providing a more general example.

Lines 62-64. Over what time frame has land use and land cover change contributed 12% and reduced NPP by 25%?

Line 69. Not every reader will know that Indonesia is a leading producer of palm oil and rubber. Consider stating this before comparing Indonesia’s forest loss to Brazil, so that this statistic makes more sense.

Lines 72-76. This is where I’d like to see more incorporation of previous studies, the data and insights they provide, and their shortcomings. As currently written, this does not do justice to the work of others on this topic (even if that work is lacking in certain ways). Also, the authors might want to state clearly why IPCC factors are insufficient (it might not be obvious to all readers).

Line 75. It seems a bit of a stretch to say that a comprehensive dataset cannot be collected by a single research group – perhaps amend to Unlikely or Difficult.

Line 92. Is smallholder jungle rubber really a plantation? I’m not convinced. Consider a different word.

Lines 130-131. “Unlike other land use types...oil palm plantations did not include CO₂ efflux from litter”. Is this because it was not measured? Or because there was no litter? Not clear.

Lines 131-132. “Considering decomposition of frond piles...” Please defend why (or why not) this is a reasonable assumption.

Line 151. “Time-averaged biomass C stocks...” Please clarify – averaged over a typical plantation lifetime? I assume this is in the methods but a short clarification here would help.

Lines 152-154. “Rubber plantations have a longer rotation time (40 y) than oil palm plantations (25 y) and also had a higher net biomass C uptake, without yield (3.1 ± 0.2 Mg C ha⁻¹ y⁻¹)” Jungle rubber also? Or just intensive rubber? Unclear.

Lines 156-172 (two paragraphs that begin with “Mature oil palm plantations...”). I think this is where the authors would need to discuss the fact that autotrophic and heterotrophic respiration fractions were not (or could not be) measured, and the implications of this for accounting. This would better prepare the reader for the hypothetical statements such as “only if soil heterotrophic CO₂ contributed to 50% or less...”, which are confusing as currently explained.

Line 176. By region, do the authors mean soil type? Also, what is RDA? Never defined this acronym.

Lines 179-180. “variables contributing more to the first principle component than their average weights...” Could this be explained this more plainly? Communicating this analysis with less jargon would help all readers understand.

Lines 182-183. Why was this fine root finding surprising?

Lines 204-209. While there is nothing incorrect about these sentences, they don't highlight the novel findings of the research. A large literature indicates a net C loss on conversion of tropical forest to tree crops, including oil palm and rubber. Consider re-focusing the beginning of the discussion on the most important finding (or highlight why these methods produce very robust results, which may be part of the study's novelty).

Lines 211-215. I don't understand. If the authors are assuming linear sequestration, why couldn't time-averaged C stocks for rubber be predicted? Plus, the manuscript does report the time averaged C stocks, right? Why not present the conversion results with the time-average assumptions?

Lines 217-221. Other hypotheses for the difference between the Germer et al. estimate and the estimate in this manuscript: different planting densities, or the linear assumption is not a good assumption. The RSPO in 2013 published a white paper (reports from the technical panels of the 2nd GHG working group of the RSPO) that compiles several oil palm biomass estimates – the authors might want to increase the scope of this discussion as the Germer et al. estimate is one of many.

Lines 221-222. “the potential for...” This is a finding that is already known, and supported by substantial literature. If the authors keep this sentence it might be good to add “..due to FOREST land clearing”, as conversion from grassland may have a different C balance.

Lines 236-238. Are these heterotrophic respiration figures from this study, or this research group, or another research effort? Please be specific as to the provenance of the information – not clear as written.

Lines 246-247. “Soil co₂ efflux started to decrease”. I'm not sure what this means. Started when? According to what data? Also, what is meant by “lost at an earlier stage”? Stage of what?

Lines 252-253. “sequestered for only a short time” – This is great. However, for the reader who does not know much about oil palm product uses, some clarification would be good – why is it only sequestered for a short time?

Line 253. “Uncertainties in autotrophic CO₂.” Please be more precise, e.g., “because we could not determine the contribution of autotrophic respiration to total co₂ flux, we could not assess...”

Line 255-259. This argument is hard to follow. Please revise.

Line 259. “Rh was in the upper range...” Are these results from this study? Not clear.

Lines 262-263. Soil erosion in oil palm plantations is substantial! Have others measured carbon export due to erosion? If so it would be great to provide some information on the likely scale of this flux.

Lines 255-257. Please clarify that these findings were from the same research project.

Lines 276-280. How does this payback time compare to those computed by other studies?

Line 281-283. "...not a valid argument...logged forests or agroforests" This is a value judgement. Please modify or remove this sentence.

Line 285. This is the first time that ecosystem services have been referenced. Consider adding a sentence to explain what is meant by regulating and provisioning ecosystem services.

Lines 288-289. "...can limit C losses and significantly increase the C efficiency of extensive plantation." This contrasts with previous statements about the efficiency of oil palm.

Line 291. The potential productivity of different species/cultivars? Please clarify.

Lines 293-294. Is there literature to support this statement about the impacts of management on yields in rubber? It would be great to add some evidence to these claims.

Lines 294-296. However, in this case the study is looking at smallholder oil palm, so I'm not sure how this claim applies.

Figure 3. fNPpt seems not to be in the graph, and Rh max should be explained in the caption.

Table 1. Why is Net Biomass C Update unavailable for jungle rubber?

Supplementary Table 2. "Plantation equilibrium" is misleading, because forest is included.

Detailed method for SOC equilibrium. "Equilibrium", to me, implies that things are balanced equally. The authors might consider using a different word for this idea – "balance" might be one option.

REFERENCES

Brauman, K. A., S. Siebert, and J. A. Foley. 2013. Improving crop water productivity increases water sustainability and food security—a global analysis. *Environmental Research Letters* 8:02403.

Carlson, K. M., J. S. Gerber, N. D. Mueller, M. Herrero, G. K. MacDonald, K. A. Brauman, P. Havlik, C. S. O'Connell, J. A. Johnson, and S. Saatchi. 2017. Greenhouse gas emissions intensity of global croplands. *Nature Climate Change*.

Koh, L. P., and J. Ghazoul. 2010. Spatially explicit scenario analysis for reconciling agricultural expansion, forest protection, and carbon conservation in Indonesia *Proc Natl Acad Sci USA* 107:11140-11144.

Mueller, N. D., J. S. Gerber, M. Johnston, D. K. Ray, N. Ramankutty, and J. A. Foley. 2012. Closing yield gaps through nutrient and water management. *Nature* 490:254-257.

Pittelkow, C. M., M. A. Adviento-Borbe, J. E. Hill, J. Six, C. van Kessel, and B. A. Linquist. 2013. Yield-scaled global warming potential of annual nitrous oxide and methane emissions from continuously flooded rice in response to nitrogen input. *Agriculture Ecosystems & Environment* 177:10-20.

van Noordwijk, M., H. Ningsih, and S. Rahayu. 2015. Carbon neutral? No change in mineral soil carbon stock under oil palm plantations derived from forest or non-forest in Indonesia. *Agriculture, Ecosystems & Environment* 211:195-206.

van Straaten, O., M. D. Corre, K. Wolf, M. Tchienkoua, E. Cuellar, R. B. Matthews, and E. Veldkamp. 2015. Conversion of lowland tropical forests to tree cash crop plantations loses up to one-half of stored soil organic carbon. *Proceedings of the National Academy of Sciences* 112:9956-9960.

Reviewer #3 (Remarks to the Author):

“Carbon costs and benefits of tropical land-use intensification” by Thomas Guillaume et al.

General comment:

I am quite positive with this study that provides a great and complete picture of C budgets of two major land-use types following forest conversion in South East Asia. This study provides a thorough assessment of the main C pools and fluxes in rubber and oil palm plantations in comparison with surrounding forests. By quantifying the Net Primary Productivity of those three land-cover types, the authors are providing a balanced view of the total C costs of each pathway. If the methods and sampling design comply with standard analyses of ecosystem C budget, additional information is required (see below). In the contrary, the discussion could be shortened to highlight a few key results. More generally, an important aspect of forest conversion lies in the dramatic collapse of key ecosystem functions (i.e. water storage, air/water filtration, timber and non-timber products) and biodiversity. I understand the authors are interested in C budget, but putting those issues in perspective of their results would certainly balance the sole carbon view-point harbored by the authors.

Major comments:

While I understand that this study builds up on previous works, a minimal background information, notably on the way above- & below-ground biomass stocks and fluxes were estimated, would ease the reading and understanding of the manuscript. Kotowska et al. (2015) did not estimate AGB fluxes and the methods for estimating NPP (C fluxes) is not detailed enough. For instance, how were the 40 trees per plot selected and re-measured? Why not estimating biomass with the most recent allometric equations proposed in Chave et al. (2014)? There is also ways to propagate uncertainties all the way through (check for the BIOMASS R package and Pearson et al. 2014). Owing the importance forest stocks and fluxes have in this study (serving as benchmark), one want to know precisely how these figures were computed. Same for AGB of oil palm, there are more recent studies and allometric models that Asani et al., notably Khasanah et al. (2015). This last study is reporting a time-average AGB stock (25 years) of 42 Mg C/ha (versus 52 in this study). Once again, discussing recent results of other studies would allow getting more contrasting view on the sole C “benefit” of

oil palm plantations. I am fully aware that using alternative allometric models will raise different results and that getting precise AGB/C stocks is not key here. However, a sensitivity analysis (through a proper error propagation) would probably be useful to understand the magnitude of differences among land-use types. I don't think that merely computing SD across plots provides a rigorous way to estimate the uncertainties surrounding both C stocks and fluxes.

I. 331: Few is said on the way fine and coarse roots were measured. How are the pits located in the plots? Obviously the concentration of roots depends on the density of stems and distance to nearest stem. Further, depending on land management in oil palm concession (i.e. application of glyphosate vs manual land clearing before harvest), there might be quite a lot of dead roots that add to oil palm root mass. Here as well some more information is required.

Finally, as oil palm plantations are very detrimental for soils, isn't it expected to have a progressive reduction in yield at either tree or concession level through time? Would it be possible to go beyond a single rotation here, making the crude assumption that nothing could succeed to oil palm plantations ?

Minor comments:

I. 268: "Since the studied forests had experienced light logging in the past, the biomass might still slightly increase , leading to a small C sink in forests(Drescher et al., 2016). I suggest to rephrase as follow: "(..) plots might still be recovering biomass, leading to a small C sink in forests" and cite a more appropriate reference such as Rutishauser et al. 2017.

I. 386 : from my own experience, rubber wood is very popular in SEA and the wood is used for various purposes. It might be meaningful to account for a fraction of wood in the total NPP exported from the ecosystem (Fig. 1).

Litterature cited:

Chave, J., Réjou-Méchain, M., Búrquez, A., Chidumayo, E., Colgan, M.S., Delitti, W.B., Duque, A., Eid, T., Fearnside, P.M., Goodman, R.C., Henry, M., Martínez-Yrizar, A., Mugasha, W.A., Muller-Landau, H.C., Mencuccini, M., Nelson, B.W., Ngomanda, A., Nogueira, E.M., Ortiz-Malavassi, E., Pélissier, R., Ploton, P., Ryan, C.M., Saldarriaga, J.G., Vieilledent, G., 2014. Improved allometric models to estimate the aboveground biomass of tropical trees. *Glob Change Biol* 20, 3177–3190. doi:10.1111/gcb.12629

Khasanah, N., van Noordwijk, M., Ningsih, H., 2015. Aboveground carbon stocks in oil palm plantations and the threshold for carbon-neutral vegetation conversion on mineral soils. *Cogent Environmental Science* 1. doi:10.1080/23311843.2015.1119964

Pearson, T., Walker, S., Brown, S., 2005. Sourcebook for land use, land-use change and forestry projects. BioCarbon Fund & Winrock International.

Rutishauser, E., Hérault, B., Baraloto, C., Blanc, L., Descroix, L., Sotta, E.D., Ferreira, J., Kanashiro, M., Mazzei, L., d'Oliveira, M.V.N., de Oliveira, L.C., Peña-Claros, M., Putz, F.E., Ruschel, A.R., Rodney, K., Roopsind, A., Shenkin, A., da Silva, K.E., de Souza, C.R., Toledo, M., Vidal, E., West, T.A.P., Wortel, V., Sist, P., 2015. Rapid tree carbon stock recovery in managed Amazonian forests. *Current Biology* 25, R787–R788. doi:10.1016/j.cub.2015.07.034

Response to referees

Reviewer 1

This manuscript presents a carbon budget analysis for three tree plantation types in Indonesia (jungle rubber, rubber monoculture, and oil palm monoculture) and compares them with the carbon budget of natural forest. The topic is timely and the question being asked here is of great societal benefit. The authors present a very thorough C budget analysis. While I see the value of this work, the manuscript needs substantial revision. I summarize my key recommendations below:

> We would like to express our gratitude to the reviewer for his overall positive feedback and his constructive comments. We revised the manuscript as suggested.

1. The title and beginning of the abstract/introductions made me think this was a global study, when in fact it is focused specifically on tree plantations in Indonesia. The title should reflect this, and the abstract and introduction could also state this up front.

> We thank the reviewer for pointing out the risk of confusion regarding the manuscript's scope from the submitted title and introduction. The manuscript presents an comprehensive analysis on the functioning and services related to C within rainforests and how they are affected by two of the main crops driving past and future land-use change in the tropics. As such, we have modified the title to reflect which land-use types the manuscript refers to. All land-use types investigated are now stated in the abstract and are mentioned in the second paragraph of the introduction.

2. The abstract is extremely vague. It needs more specifics. I give examples in my detailed comments below.

The abstract in general is overly vague. I recommend adding in quantitative information whenever possible, for example how much forested land (e.g., in hectares) does Intensive Oil Palm allow to remain forested? How unbalanced is the sharing of NPP? Strongly perturbs the functioning (what functions?), what other services?

Efficient to produce yield? Net primary production loss for ecosystem functioning? I think this sentence needs rephrasing so that it is clear. Yield of what?

> We agree that the abstract was too vague and did not adequately reflect our most striking findings. We have added numbers to specify the total C losses and to stress the importance of belowground C losses. Because of the word limit we haven't specified all numbers associated with the information given in the abstract, but we have added a new figure (Fig. 3) to show the unbalanced share of NPP so that the information can be easily found. We have also specified that we address regulating ecosystem services. As suggested by reviewer 2, we remove the topic about that oil palm potentially sparing forest as compared to other crops.

The focus of the paper appears to be global, so the discussion of research plots in Indonesia is confusing here.

> We believe that the fact that the information that all data were collected in the same research plots is part of the manuscript's novelty and mentioning the geographical area that is at the forefront of tropical land-use intensification reinforces the relevance of the findings. As such, we have kept the geographical mention in the abstract but have moved the methodological aspect at the end of the discussion.

Efficient to produce yield? Net primary production loss for ecosystem functioning? I think this sentence needs rephrasing so that it is clear. Yield of what?

> We have rephrased his part of the abstract

3. I recommend including more background on the plantations studied, how they are managed, what do you mean by more intense management? Why did you choose these sites to address this larger question? Discuss net primary productivity and other aspects to help the audience follow the logic of the paper. This would help prepare the audience for the very detailed C budget review in the rest of the paper.

As a whole, more introduction into the plantation types studied is needed. Why were these land use types chosen? How are they managed?

> We have modified the structure of the introduction. Information relating to NPP is now included in the 3rd paragraph. Information relating to C stocks, previous findings and challenges are now presented in the 4th paragraph. Information on land-use intensity and the relevance of the sites is now included in the 5th paragraph.

Additional details on sites and management are now added in the methods section.

4. As a whole, I had a very difficult time following the paper. The authors present a LOT of information, a conceptual diagram showing the C pools analyzed would help. The introduction could also provide a road map to help lead the audience through the details of all of the analyses.

I also wanted more introduction into the specific measurements being made. It is overwhelming to get to the methods section without a sense of all of the considerations that went into measuring the C pools and fluxes. What is it that makes this study unique?

> Reviewers 1 and 2 have highlighted the writing clarity as a key aspect to improve the quality of the manuscript. To improve it, we have defined four lines of evidence from the aims and have followed these throughout the rest of the text, i.e. 1) overall C stocks, 2) pools taken separately starting from aboveground to belowground, 3) C fluxes and their balance and 4) tradeoffs. All measurements are now mentioned before the aims, alongside a brief description of the methodology and the overall study design. Figure 2 now presents all measured pools and fluxes together and shows the relationships between them. Because the figure is referenced in the first paragraph of the results section, we believe that an additional conceptual figure in the introduction would now be redundant with Figure 2. However, we have added an additional panel to Figure 4 (previously Figure 3) that explains which flux measurements (input vs output) must be compared to assess net ecosystem productivity, ecosystem C storage and soil C storage.

5. I think that the authors need to be very careful with how they are using different terms. For example, NPP is a rate, but the authors discuss NPP as if it were a pool of Carbon. It is important to be precise with language and to ensure that key terms are defined. NPP isn't "removed" from an ecosystem, the biomass is removed from the system, and then productivity declines.

NPP is not a pool, it is a rate (e.g., Mg C/ha/year). Litterfall is a product of NPP (e.g., C fixed from the atmosphere). We use it as an indirect measure of above ground productivity. If litterfall is removed from a system, biomass is being removed, not NPP.

The use of NPP is not quite right. This sounds to me like the authors are discussing carbon stocks and not fluxes.

> We thank the reviewer for pointing out this semantic confusion. We have replaced expressions such as yield, NPP exported or remaining by harvested biomass, produced biomass, or other expression referring to biomass rather than its production rate.

6. In the NEP section, the authors do not account for woody respiration. This should be acknowledged. This is why definition of terms is really important. Some of the measurements made are indirect measures of NPP and NEP and the assumptions being made need to be clearly spelled out.

L418 the authors are not accounting for woody respiration, which should be acknowledged and justified.

> There is CO₂ emitted during wood decomposition in jungle rubber and forests but not in rubber and oil palm monocultures because of the absence of dead wood. Our approach took advantage of this feature specific to monocultures but was relying on assumptions for jungle rubber and forest. Therefore, we have removed the estimation for jungle rubber and forest from the revised version (Figure 4, previously 3).

Nonetheless, we have kept in Supplementary Table 2 the estimation of the fraction of heterotrophic respiration in soil CO₂ efflux in forest under the assumption that forest pools are at equilibrium. Following this assumption, subtracting wood production from the total NPP compensates for not accounting for wood decomposition. This value shows that ecosystem C storage in rubber monocultures does not increase unless the autotrophic respiration is similar or higher than in forest. All methodological details appear now in the methods section and supplementary information.

7. There are quite a few grammatical errors in the paper that should be addressed.

> We apologize for these errors. A native English speaker has checked the revised version.

L52 “increased” greenhouse gas emissions

> Done

L64 This is confusing. NPP isn't “available” for ecosystem function, it is an ecosystem function.

> Sentence removed

L76 a citation is needed here

L76 can you quantify how much data are default versus real values? And are the defaults used from tropical systems?

> We have modified the sentence to highlight the limitations of the Tier 1 methodology without referring to specific studies and have added a citation. Yes default values exist for each ecological zone and land-use type as defined by the IPCC.

L74 I disagree with this statement. There are research groups that perform whole system C budgets. They are often scarce because some numbers are difficult to get e.g., belowground pools can be difficult to quantify.

> Modified to “are difficult to conduct by a single research group”

L75 spell out IPCC the first time

> Done

L77 please spell out the acronym used here

> Done

L80 what is “plantation scale” do you mean by studying individual plantations?

> Modified to “within plantation boundaries”. We intended to express that the study does not account for externalities (e.g. C emissions from fertilizer production), nor does it upscale the impacts to landscape or global scales.

L87 “reduced NPP remaining in the system” this needs to be rephrased.

> Done

L178 yield of what? What is being harvested exactly?

> We have added in the results section that the harvested biomass is in the form of fresh fruit bunches in oil palm plantations and latex in rubber plantations.

L212 up “to” 40

> Done

L213 “on” average

> Done

L215 “On” the other hand

> Done

The tree species should be listed here rather than using the common names.

> Latin names were added when common names appear for the first time in the introduction and methods sections

L343 how many jungle rubber plantations were studied?

> All land-use types had 8 plantation replicates. We also added this information before the results section

Reviewer 2

This manuscript uses an extensive dataset on carbon stocks and fluxes across four land uses with high tree cover in Sumatra, Indonesia to evaluate how land use affects carbon storage dynamics in the ecosystem, as well as harvested carbon yields. The authors find that while oil palm has the most net carbon loss (compared to natural forest), it also yields the most carbon for human use. The work is of interest to academics and practitioners working in tropical land use change, and may inform design of land-based carbon projects (e.g., REDD+) [although the authors do not engage with this policy angle in their paper]. However, I had a hard time understanding the big novel finding from the work that would move the field forward, especially given that this group has previously published some of these data, including in Nature Communications (doi:10.1038/ncomms13137). Perhaps the most novel finding, in my opinion, is that soil organic carbon in these mineral soils appears to still be decreasing in oil palm plantations. The statistical analysis seems robust, and

the authors provide detailed (for the most part) methods documenting their procedures that should allow for reproduction of these results if required. Beyond the issue of novelty, there are some limitations of the manuscript as currently written. I outline these general issues and specific comments below.

> We would like to thank the reviewer for pointing out the weaknesses of the previous version of this manuscript and for his effort to give us valuable advices to solve them. Following his feedback, we have adjusted the text, figures and references to improve the clarity and to highlight novelties as advised.

GENERAL COMMENTS

1. Clarity. In reading the manuscript, I often found that I was confused. The biggest improvement that the authors can make to the article, in my opinion, is in clarity of writing.

> Reviewers 1 and 2 have highlighted the writing clarity as a key aspect to improve the quality of the manuscript. To improve it, we have defined four lines of evidence from the aims and have followed these throughout the rest of the text, i.e. 1) overall C stocks, 2) pools taken separately starting from aboveground to belowground, 3) C fluxes and their balance and 4) tradeoffs. All measurements are now mentioned before the aims, alongside a brief description of the methodology and the overall study design. Figure 2 now presents all measured pools and fluxes together and shows the relationships between them. Because the figure is referenced in the first paragraph of the results section, we believe that an additional conceptual figure in the introduction would now be redundant with Figure 2. However, we have added an additional panel to Figure 4 (previously Figure 3) that explains which flux measurements (input vs output) must be compared to assess net ecosystem productivity, ecosystem C storage and soil C storage.

One source of confusion was the difference between data that was collected/analyzed for this article, versus previous findings by the group, versus findings of others. Placing these in context (e.g., “previous research in these study sites reported”... or “in Kalimantan, XX author found that”) would help immensely.

> To address this issue, we have proceeded as follow. First we now mention at the end of the introduction which stocks and fluxes were previously published and synthesized in this manuscript, and also which previous data were updated with new allometric equations. When results from single pools or fluxes are presented for the first time, i.e. findings could be directly inferred from the initial publication, we have now added the reference. Data sources are now summarized in Supplementary Table 1. Finally, findings on our research sites that are not included in the dataset (e.g. not available for all sites or not on C) are now mentioned using “A previous research on our study sites...”. It happened once that the finding was mentioned in the results section. Thus, we moved it to the discussion section.

In addition, only after I read the entire manuscript did it become clear that the authors did not measure heterotrophic respiration, but infer the heterotrophic/autotrophic partition. This is fine, but stating this clearly in the main text of the article is needed so that much of the results and discussion make sense!

> We fully agree that this information was not stated early enough in the manuscript. It now first appears in the results section. The presentation of the C storage approach in the results and discussion sections and in Figure 4 (previously 3) has been deeply modified in the revised version to improve its clarity. Details are explained in

comments below.

2. Novelty. This manuscript brings together a huge amount of robustly-collected data to understand carbon fluxes in a region of Sumatra. However, the authors don't do a great job of highlighting what is novel about this work, especially compared to their previous publication in Nature Communications that also reports NPP, harvested biomass, and carbon stocks. In the introduction, the authors imply that there has not been region-specific accounting of carbon fluxes under land use change to oil palm and rubber. There are many studies that tackle this problem, but these other studies are mostly not acknowledged by the authors, nor are their shortcomings (and they have shortcomings!) discussed. The authors might want to grapple specifically with the contrasting findings from (van Noordwijk et al. 2015) and (van Straaten et al. 2015), or the C measurements of (Koh and Ghazoul 2010) – I'm sure there are others too. This would better highlight the novelty of the work, and its contribution not just to "advising tropical countries in their development policies", but also to the theoretical and empirical understanding of carbon dynamics in tropical tree plantations. Personally, I think that the contribution to understanding soil C stock dynamics (including inputs and removals) as well as the partitioning among yield versus ecosystem NPP is fascinating, but the authors should determine what they think is novel about their manuscript, in the context of other literature.

> We share a similar viewpoint on the novelties of the manuscript and the reasons why they were not made apparent enough in the previous version. Because of the large scope of previous article in Nature Communications (mainly standardized data to illustrate tradeoffs between various socio-economic and environmental aspects), many fundamental aspects related to the C cycle within these agroecosystems were not addressed. Not all pools were accounted for (especially dead biomass), and stocks were not aggregated or compared among pools to determine their relative sensitivity to land-use change. By focusing on the C cycle in this study, we were able to quantify the overall impacts on ecosystem C storage, underline the importance of belowground C losses and to highlight the strong imbalance in the share of NPP, alongside discussing its effects on ecosystem processes and SOC dynamics. To emphasize these novelties, we have followed the suggestion of this reviewer to introduce and discuss more findings and knowledge gaps highlighted by previous studies. Furthermore, we have added a new figure (Fig. 3) to illustrate the share of NPP and the decrease of C available for the decomposer food web. Finally, we have changed the structure and focus of the introduction and discussion.

3. Land Sparing. In the abstract, the authors claim that oil palm spares forested land. This statement is not supported by the manuscript, and is highly contextual (e.g., see recent review by C. Kremen in the Annals of the New York Academy of Sciences, as well as Carrasco et al. paper on increasing oil palm yields in Science). I would revise the abstract to avoid grappling with this sticky topic.

> This topic has been removed from the manuscript

4. Justification for Statistical Approach. While I think the statistical approach is likely a good choice, I'd like to see a justification for these tools in the methods – just a few sentences will ground the methods and argue for why they are appropriate to the data at hand.

> We have complemented the subsection "statistics and calculations" with an explanation of when Kruskal-Wallis or Wilcoxon tests were used, as well as the

calculation methodology for the error associated with the comparison of two land uses (e.g. C losses).

SPECIFIC COMMENTS

Abstract. In the abstract, it would be great to see more results from the paper, especially novel findings or specific values. As currently written, the abstract contains just one sentence describing findings from this research.

> According to the suggestions of reviewers 1 and 2, we have modified the abstract to include more specific findings.

Line 56. “The tropics has a lower eco-efficiency”. It seems that the tropics *could* have high yields, and in some cases, they do (how about soy in Brazil?). Also, there are other ways to measure these tradeoffs beyond just carbon. See e.g. (Pittelkow et al. 2013), (Mueller et al. 2012), (Brauman et al. 2013), (Carlson et al. 2017) - perhaps the authors need to define eco-efficiency more specifically related to carbon.

> We have specified in this sentence about which eco-efficiency we are referring to.

Line 60. “phytosanitary products for oil palm cultivation” – this is the first time in the main text that oil palm has been introduced, so it feels uber-specific – consider providing a more general example.

> We agree with the reviewer on the importance of introducing early which land-use types the manuscript is focusing on and their relevance. The structure of the introduction was modified accordingly.

Lines 62-64. Over what time frame has land use and land cover change contributed 12% and reduced NPP by 25%?

> Sentence has been removed from the revised version.

Line 69. Not every reader will know that Indonesia is a leading producer of palm oil and rubber. Consider stating this before comparing Indonesia’s forest loss to Brazil, so that this statistic makes more sense.

> Done

Lines 72-76. This is where I’d like to see more incorporation of previous studies, the data and insights they provide, and their shortcomings. As currently written, this does not do justice to the work of others on this topic (even if that work is lacking in certain ways). Also, the authors might want to state clearly why IPCC factors are insufficient (it might not be obvious to all readers).

> We have incorporated this suggestion into the 4th paragraph of the introduction. We now introduce the uncertainties surrounding the impact of forest conversion on belowground C stocks (referring to previous research) and the fact that SOC losses are not accounted for in the main methodologies developed to assess greenhouse gas emissions from these land-use changes or the sustainability of new plantation developments.

Line 75. It seems a bit of a stretch to say that a comprehensive dataset cannot be collected by a single research group – perhaps amend to Unlikely or Difficult.

> Modified to “are difficult to conduct by a single research group”

Line 92. Is smallholder jungle rubber really a plantation? I’m not convinced. Consider

a different word.

> From our understanding, the definition of plantation does not exclude agroforest or other cultivation types that are less intensive than monocultures. We used plantation as a generic word when trees are planted and thus use this term when a result applied to any of the three agroecosystems. When we needed to make a distinction between the types of cultivation, we have opposed monocultures to agroforests. We could replace plantations with agroecosystems but we have decided that this is too broad.

Lines 130-131. “Unlike other land use types...oil palm plantations did not include CO₂ efflux from litter”. Is this because it was not measured? Or because there was no litter? Not clear.

> No chambers were located in frond piles, which occupy about 15% of the plantation surfaces, even though their location was chosen randomly. The results, discussion, methods and SI dealing with the flux balances have been modified accordingly.

Lines 131-132. “Considering decomposition of frond piles...” Please defend why (or why not) this is a reasonable assumption.

> We have modified the text and added references to explain that frond pile C stocks are constant in mature OP and that frond C stabilization in SOC is negligible compared to frond production.

Line 151. “Time-averaged biomass C stocks...” Please clarify – averaged over a typical plantation lifetime? I assume this is in the methods but a short clarification here would help.

> Done

Lines 152-154. “Rubber plantations have a longer rotation time (40 y) than oil palm plantations (25 y) and also had a higher net biomass C uptake, without yield (3.1 ± 0.2 Mg C ha⁻¹ y⁻¹)” Jungle rubber also? Or just intensive rubber? Unclear.

> We have specified “rubber monocultures”.

Lines 156-172 (two paragraphs that begin with “Mature oil palm plantations...”). I think this is where the authors would need to discuss the fact that autotrophic and heterotrophic respiration fractions were not (or could not be) measured, and the implications of this for accounting. This would better prepare the reader for the hypothetical statements such as “only if soil heterotrophic CO₂ contributed to 50% or less...”, which are confusing as currently explained.

> The paragraph has been modified. We now introduce the reader to the approach, to the fact that heterotrophic respiration was not measured and how we overcome it.

Line 176. By region, do the authors mean soil type? Also, what is RDA? Never defined this acronym.

> We have specified that the design was replicated in two regions. Details are in the methods section. We have specified the acronym for RDA and the constraining factors (redundancy analysis; similar to a PCA but constrained by explanatory factors).

Lines 179-180. “variables contributing more to the first principle component than their average weights...” Could this be explained this more plainly? Communicating this analysis with less jargon would help all readers understand.

> Paragraph modified accordingly

Lines 182-183. Why was this fine root finding surprising?

> We have removed surprising and have reformulated the sentence to express that fine root variables do not follow the same trends as other biomass variables.

Lines 204-209. While there is nothing incorrect about these sentences, they don't highlight the novel findings of the research. A large literature indicates a net C loss on conversion of tropical forest to tree crops, including oil palm and rubber. Consider re-focusing the beginning of the discussion on the most important finding (or highlight why these methods produce very robust results, which may be part of the study's novelty).

> We have now started the discussion by highlighting the importance of belowground SOC losses and the inconsistency of default biomass values used by methodologies assessing land-use change greenhouse gas emissions or plantation sustainability with our results and previous findings.

Lines 211-215. I don't understand. If the authors are assuming linear sequestration, why couldn't time-averaged C stocks for rubber be predicted? Plus, the manuscript does report the time averaged C stocks, right? Why not present the conversion results with the time-average assumptions?

> We have specified that time-averaged C stocks were not estimated only for jungle rubber and why it was not appropriate for this land use type. We have added C losses for monocultures calculated with time-averaged biomass C stocks in Table 1 and now discuss them. We chose not to present them as main results, for instance in Figure 1, because 1) data are not available for all land-uses and pools (e.g. jungle rubber or SOC) and 2) estimates depend strongly on the rotation time. This varies if farmers shorten or extend it and so time-averaged C stocks are potential, not actual stocks. Nonetheless, we chose to show the range of C efficiencies calculated with time-averaged C stocks to compare the potential of monocultures. Finally, SOC stocks dynamics do not follow rotation cycles. We would have to extrapolate the number of plantation cycles that will occur to calculate a time-average SOC stocks.

Lines 217-221. Other hypotheses for the difference between the Germer et al. estimate and the estimate in this manuscript: different planting densities, or the linear assumption is not a good assumption. The RSPO in 2013 published a white paper (reports from the technical panels of the 2nd GHG working group of the RSPO) that compiles several oil palm biomass estimates – the authors might want to increase the scope of this discussion as the Germer et al. estimate is one of many.

> We have changed this part of the discussion and have compared the updated data with more studies.

Lines 221-222. “the potential for...” This is a finding that is already known, and supported by substantial literature. If the authors keep this sentence it might be good to add “..due to FOREST land clearing”, as conversion from grassland may have a different C balance.

> This topic has been removed from the discussion.

Lines 236-238. Are these heterotrophic respiration figures from this study, or this research group, or another research effort? Please be specific as to the provenance of

the information – not clear as written.

> We now mention clearly the source of the data.

Lines 246-247. “Soil co2 efflux started to decrease”. I’m not sure what this means. Started when? According to what data? Also, what is meant by “lost at an earlier stage”? Stage of what?

> This topic has been removed from the discussion.

Lines 252-253. “sequestered for only a short time” – This is great. However, for the reader who does not know much about oil palm product uses, some clarification would be good – why is it only sequestered for a short time?

> Information is now available in the text.

Line 253. “Uncertainties in autotrophic CO2.” Please be more precise, e.g., “because we could not determine the contribution of autotrophic respiration to total co2 flux, we could not assess...”

> Done

Line 255-259. This argument is hard to follow. Please revise.

> Removed from discussion

Line 259. “Rh was in the upper range...” Are these results from this study? Not clear.

> Removed from discussion

Lines 262-263. Soil erosion in oil palm plantations is substantial! Have others measured carbon export due to erosion? If so it would be great to provide some information on the likely scale of this flux.

> We have now provided an estimation of such a scale by multiplying soil erosion rates from a study in Malaysia with the average C content on our sites

Lines 255-257 -> 265-267?. Please clarify that these findings were from the same research project.

> Done

Lines 276-280. How does this payback time compare to those computed by other studies?

> Removed from discussion

Line 281-283. “...not a valid argument...logged forests or agroforests” This is a value judgement. Please modify or remove this sentence.

> Removed from discussion

Line 285. This is the first time that ecosystem services have been referenced.

Consider adding a sentence to explain what is meant by regulating and provisioning ecosystem services.

> Done

Lines 288-289. “...can limit C losses and significantly increase the C efficiency of extensive plantation.” This contrasts with previous statements about the efficiency of oil palm.

> Indeed, this was a mistake. We have removed “and significantly increase the C efficiency of extensive plantation”

Line 291. The potential productivity of different species/cultivars? Please clarify.
> Removed from discussion

Lines 293-294. Is there literature to support this statement about the impacts of management on yields in rubber? It would be great to add some evidence to these claims.
> Removed from discussion

Lines 294-296. However, in this case the study is looking at smallholder oil palm, so I’m not sure how this claim applies.
> Removed from discussion

Figure 3. fNPpt seems not to be in the graph, and Rh max should be explained in the caption.
> Figure and caption modified

Table 1. Why is Net Biomass C Update unavailable for jungle rubber?
> Biomass in jungle rubber follows the same dynamics as that in monocultures. The land is not fully cleared of vegetation before planting rubber trees. The biomass depends mostly on the management of standing biomass of native trees by the farmer rather than on time. This variability is captured by the field replicates. We have added this explanation to the manuscript.

Supplementary Table 2. “Plantation equilibrium” is misleading, because forest is included.

> Changed to *ecosystem*

Detailed method for SOC equilibrium. “Equilibrium”, to me, implies that things are balanced equally. The authors might consider using a different word for this idea – “balance” might be one option.

> Changed to *balance*

Reviewer 3

I am quite positive with this study that provides a great and complete picture of C budgets of two major land-use types following forest conversion in South East Asia. This study provides a thorough assessment of the main C pools and fluxes in rubber and oil palm plantations in comparison with surrounding forests. By quantifying the Net Primary Productivity of those three land-cover types, the authors are providing a balanced view of the total C costs of each pathway. If the methods and sampling design comply with standard analyses of ecosystem C budget, additional information is required (see below). In the contrary, the discussion could be shortened to highlight a few key results. More generally, an important aspect of forest conversion lies in the dramatic collapse of key ecosystem functions (i.e. water storage, air/water filtration, timber and non-timber products) and biodiversity. I understand the authors are interested in C budget, but putting those issues in perspective of their results would certainly balance the sole carbon view-point harbored by the authors.

> We would like to thank the reviewer for their constructive comments and their positive feedback overall. We agree with most suggestions, and have adjusted the

manuscript accordingly. The discussion section has now been reformulated to highlight the key results more clearly and we now integrate previous findings into the discussion. We have also complemented the paragraph discussing impacts on other important ecosystem functions indirectly linked to C, such as water regulation, nutrient leaching and soil microbial and faunal communities. We agree that deforestation has broader impacts than only on the C viewpoint and we have explicitly stated this in the discussion. However, we have also avoided making an exhaustive review in this manuscript. The integration of all these suggestions has increased the discussion length slightly. Nevertheless, we are convinced that it has been significantly improved by highlighting the novelties, integrating previous findings and introducing a clearer structure overall.

Major comments:

While I understand that this study builds up on previous works, a minimal background information, notably on the way above- & below-ground biomass stocks and fluxes were estimated, would ease the reading and understanding of the manuscript. Kotowska et al. (2015) did not estimate AGB fluxes and the methods for estimating NPP (C fluxes) is not detailed enough. For instance, how were the 40 trees per plot selected and re-measured?

> We have added all requested details to the methods section and have expressed more clearly that AGB fluxes were also measured by Kotowska et al. (2015).

Why not estimating biomass with the most recent allometric equations proposed in Chave et al. (2014)? There is also ways to propagate uncertainties all the way through (check for the BIOMASS R package and Pearson et al. 2014). Owing the importance forest stocks and fluxes have in this study (serving as benchmark), one want to know precisely how these figures were computed. Same for AGB of oil palm, there are more recent studies and allometric models that Asani et al., notably Khasanah et al. (2015). This last study is reporting a time-average AGB stock (25 years) of 42 Mg C/ha (versus 52 in this study). Once again, discussing recent results of other studies would allow getting more contrasting view on the sole C “benefit” of oil palm plantations. I am fully aware that using alternative allometric models will raise different results and that getting precise AGB/C stocks is not key here.

> We thank the reviewer for this suggestion. We have now updated the initial values published by Kotowska et al. (2015) with the allometric equations published by Chave et al. (2014) and the power regression model on mineral soil from Khasanah et al. (2015). The equation published by Chave et al. (2014) changed the estimations only marginally but the equation published by Khasanah et al. (2015) increased our oil palm biomass estimation, up to the point that rubber and oil palm now possess similar AGB. Originally, we used the modified equation of Asari et al. (2013) for oil palm, which is based on a destructive sampling carried out by Khalid & Anderson 1999 and allows us to estimate individual palm biomass based on palm height. To validate the equation, we harvested 3 mature oil palms from our research area and measured dry mass. We agree that this method is less robust than that used by Khasanah. Furthermore, it was estimating low AGB values compared to literature values. Therefore, we believe that the updated values are accurate and that they justify adapting all the calculations and figures in the manuscript

However, a sensitivity analysis (through a proper error propagation) would probably

be useful to understand the magnitude of differences among land-use types. I don't think that merely computing SD across plots provides a rigorous way to estimate the uncertainties surrounding both C stocks and fluxes.

> Following the comment of the reviewer, we corrected a mistake in the error associated with C loss estimates. The error associated with the reference land-use, i.e. forest, was not included, leading to an underestimation of the error. The new error was calculated using the error propagation theory as following:

$$\Delta C \text{ losses} = \sqrt{(\Delta \text{forest})^2 + (\Delta \text{plantation})^2}$$

The standard error (SE) computed across the 8 land-use replicates is the sum of all random errors, i.e. 1) random measurement errors (imprecision in measures), 2) random sampling errors at the plot scale (natural plot heterogeneity) and 3) random sampling error between plots (natural land-use variability). We agree with the reviewer that the presented SEs don't enable us to disentangle the contribution of each type of error and overestimate the natural variation of the "population" of each land-use type. The biomass R package is an interesting tool to assess the random error associated with the AGB measurements, especially when default values are used for tree height, tree diameter and wood density. Since we measured these 3 parameters, the package is useful to assess the measurement error associated with the error of allometric parameters. This allows us to better estimate the contribution of measurement errors in the total random error for AGB. Nonetheless, not all measurement errors would still be taken into account (e.g. tree height measurement, representativeness of sampled trees, measurements of biomass C content from elemental analyzer, etc.). We haven't performed a variance partitioning analysis because it is too complex to determine all the potential sources of measurement error. We focused to have the most precise estimate of the mean for each land-use by limiting random measurement and sampling error when collecting data and the most accurate estimate of the mean by updating biomass estimates with new allometric equations (avoiding a systematic error on the mean). We have compared the SE calculated with 1) the method adopted in the manuscript consisting of summing all pools in each replicate plot and computing the SE for the mean of the 8 plots and 2) the method of computing the mean of each pool in the 8 replicate plots and summing the means of all pool to obtain total C stocks in each land-use, i.e. using the error propagation theory:

$$\Delta \text{ total C stocks} = \sqrt{\sum_i^{n=\text{number of pools}} (\Delta P_i)^2}$$

Both methods yielded similar SE:

Land-use	Mean of pool-> sum of pools	Sum of pools-> mean
Forest	4.3	4.3
JR	6.2	5.2
Rubber	9.5	9.2
Oil palm	5.9	6.8

We prefer the method 1) to avoid dealing with the assumption of variable independency required for method 2).

l. 331: Few is said on the way fine and coarse roots were measured. How are the pits located in the plots? Obviously the concentration of roots depends on the density of stems and distance to nearest stem. Further, depending on land management in oil palm concession (i.e. application of glyphosate vs manual land clearing before harvest), there might be quite a lot of dead roots that add to oil palm root mass. Here as well some more information is required.

> To clarify the methods for fine root estimates, we have added further detail to the methods section:

“Fine root biomass (diameter \leq 2 mm) was measured using 10 soil cores (3.5 cm in diameter, 50 cm soil depth) on each plot which were located in a randomly placed grid. In oil palm plantations a higher proportion of root necromass can be expected due to management such as glyphosate application, pronounced seasonality in fine root mortality (Kotowska et al. 2015b) and slower decomposition rates (Violita 2014). Therefore, all fine root segments $>$ 1 cm lengths were extracted by washing on a sieve and separated under a stereomicroscope into live (biomass) and dead (necromass) fractions. Alteration in periderm colour, non-turgid cortex, root elasticity and the absence of living root tips were used as determinants for root death.”

The reviewer is right that previous studies show a dependency of root density on the distance to the nearest tree (e.g. Henson & Chai 1997). However, we have tested this relationship and found only a marginally significant effects in oil palm plantations (Saner et al. 2015, data not shown). As we have used a randomly computed sampling grid of 10 sample points in each plot, we expect to yield reliable and unbiased results. The same grid pattern was used on each plot and was placed in an angle with the oil palm planting lines, which allowed for varying distances to the oil palm trees.

Finally, as oil palm plantations are very detrimental for soils, isn't it expected to have a progressive reduction in yield at either tree or concession level through time? Would it be possible to go beyond a single rotation here, making the crude assumption that nothing could succeed to oil palm plantations?

> As we don't have reliable data on the decrease of yield after several succession cycles, we can only speculate on this topic. In the study region, all investigated oil palm plantations were still in their first rotation. However, we personally observed plantations at the end of a second cycle (up to 56 years old) that were still fairly productive. Therefore, the assumption that nothing can succeed oil palm seems too strong, but the assumption that the yield will decrease is likely if the fertilization is not increased. We briefly mentioned the risk of this at the end on the manuscript.

Minor comments:

l. 268: “Since the studied forests had experienced light logging in the past, the biomass might still slightly increase, leading to a small C sink in forests (Drescher et al., 2016). I suggest to rephrase as follow: “(..) plots might still be recovering biomass, leading to a small C sink in forests” and cite a more appropriate reference such as Rutishauser et al. 2017.

> This discussion has been removed from the manuscript.

l. 386 : from my own experience, rubber wood is very popular in SEA and the wood is used for various purposes. It might be meaningful to account for a fraction of wood in the total NPP exported from the ecosystem (Fig. 1).

> We are aware that rubber wood can be used in construction and other permanent usages. In our study region however, rubber wood collected from plantations was mostly used as firewood. We don't have data or personal communications about wood usage at the end of the rotation cycle, but we have included the question of the fate of rubber wood in Figure 3 and in the discussion, mentioning how it would affect the C efficiency of rubber monocultures.

REVIEWERS' COMMENTS:

Reviewer #2 (Remarks to the Author):

Thank you for the very responsive revisions that have significantly improved the manuscript. I have a few more specific comments that should be addressed before publication. In addition, I remind the authors that not all reviewers are male – so the use of “he” is not recommended in responses to referees.

SPECIFIC COMMENTS

P3L40. More clear: “a decade after initial conversion” instead of “after a decade”.

P3L40. Watch your tense here and throughout, it seems that the tense switches between past and present.

P3L41. It would be great if you could specify the efficiency differences (or absolute efficiency) of the three production systems in the abstract.

P3L44. In the interest of acknowledging uncertainty, consider “[could] compromise the ability of...”. Also may want to provide a couple of examples of regulating ecosystem services for clarity.

P4L57-58. “37% of them” should be “with 37% of total area”

P4L71. Should be “deliver” not “delivers”

P4L72-74. This sentence is not clear. Please more clearly explain how a high percent of biomass harvest affects energy flow.

P6L117. “with 102-131” is unclear – is this how much carbon was lost? What does the range represent?

P6L119. From reading the methods I see that this is SE. Please note that this is standard error here (the first time presented), as well.

P6L120. Should be “rubber, rubber [and] oil palm plantations” to match syntax of carbon stocks presented previously. Also, are these values time-averaged or for one point in time? Should be clear here even if this information is provided in the methods.

P7L123. +2.7% - percent of what?

P7L125. Do the carbon figures presented above use this updated allometric equation or not? If not, why not? Not clear.

P7L126. What differences? Are these non-significant differences based on the Khasanah et al. equations or something else?

P7L127-132. This section about belowground C is confusing. I think there needs to be a bit more background about what went into these measurements and how they were done. Even though this is presented in the Methods section, short summaries of methods in the Results section will help ground your reader and illustrate how you generated the results.

P7L133. (16-28 Mg C ha⁻¹) – is this C loss?

P7L139-140. “relative inertia of SOC stocks to land use change” does not make sense. Consider: “because soil carbon stocks respond more slowly to changes in land use compared to aboveground carbon stocks” or similar.

P7L150. Please define in the Results section the threshold you are using to determine significance – if there is room it would be great to report p-values.

P10L191. “previously reported” is confusing – do you mean previously reported in this manuscript, or previously reported in another report? If you mean this manuscript, consider “described above” to clarify.

P13L277. HCS has not yet been defined as an acronym – please define the first time it is presented. More importantly, the HCSA no longer uses specific carbon thresholds but is more focused on forest structure. Please revise based on the new HCSA Toolkit.

P14L287-289. This first sentence is confusing. If calculations just use AGB, and AGB decreases, plantations would be considered C sources, right? Please clarify.

P14L289-291. Please revise to be clear that this study does not provide this answer because of the unknown heterotrophic soil respiration fraction. Notably, is probably okay to use “our” and “we” to clarify what was done in this study. As written, it remains less than clear that this was the C budgeting approach taken by these authors. Instead, it sounds like a general approach.

P14L301. “because it implies” – what implies? Revise this sentence for clarity in causality.

P15L302. “Monocultures [are] subject to”??

P16L345. “Oil palm cultivation led to [the] smallest...”

P17L360. “NPP decreases in the”

P18L378. “A previous research” is incorrect English grammar here and throughout. Should just be “Previous research”

P18L380-382. I don’t understand this connection. Is fine root biomass associated with greater water and nutrient demand? You might want to make this clear, instead of assuming that your readers know this.

P19L400-401. Most if not all intensified systems rely on external inputs. This argument seems to be beyond the scope of this article. Consider revising to focus more specifically on humid tropical plantation systems.

P26L553-555. "Differences..." – this is not a complete sentence.

Figure 1. Consider revising so that the figure is colorblind safe – this version will be a problem for people with red-green colorblindness.

Figure 2 (L747). Is the top central figure just a legend? Not clear.

Reviewer #3 (Remarks to the Author):

The authors have elegantly addressed most issues raised by the reviewers. The ms has gained in clarity, even if I would sharply shorten the discussion. There are still a few minor points that need to be solved. Please see my comments in the pdf attached.

Nice piece of work, congrats!

Ervan Rutishauser

Reviewer #4 (Remarks to the Author):

The authors present a synthesis of data (mostly previously published) in the form of a carbon budget to evaluate the impacts of rainforest conversion to rubber tree and oil palm plantations. Overall, the authors have done a great job addressing the concerns of the previous reviewers' comments. The revised manuscript is well-written and appropriate for Nature Communications. The authors seem to have refined previously confusing text in the introduction and methods, and clarified the novel aspects of their work. Figures 1, 2, and 3, in particular, are good visualizations of the conceptual framework and underlying data.

Below are a few minor comments that can help improve the manuscript further.

Minor Comments:

Ln 37 & Ln 386 - 388: I don't think it is necessary to refer to "multidisciplinary" datasets. Ultimately, this work is an ecosystem carbon budget. There are multiple data components (e.g. aboveground,

belowground, soil respiration, etc), but I would not characterize that as multidisciplinary datasets. NEP is not a multidisciplinary approach.

Ln 46 – 50: Agricultural expansion and intensification are two different phenomenon but are used interchangeably here.

Ln 75 – 86: This paragraph is a nice discussion of an important gap in our understanding and accounting of land use effects on carbon, soil organic carbon. The discussion (e.g. Ln 315 – 322) circles back to this concept nicely.

Ln 98 – 104: A little more methodological detail would be useful. For example, what were the soil depth intervals, and how many times were soils analyzed for carbon? How often were soil CO₂ and CH₄ flux measurements made?

Ln 121 – 125: It is more accurate to say “Aboveground biomass stock estimates...” The updated allometric equation did not change the stock itself, but rather the authors’ calculated estimate of the stock.

Ln 129 – 132: This is a confusing sentence. Did the plantations in the loamy Acrisol regions experience SOC loss compared to the other region with a clayey Acrisols, or compared to the rainforest? If soil type is a key variable, why was soil type ignored in the statistical analysis?

Ln 168 – 170: How are the dead oil palm frond piles managed? Are they left to passively decompose or are they composted aerobically?

Ln 360: remove the word “in”

Ln 421 – 422: Do you mean “...had not been fertilized since the year of plots establishment”?

Ln 472 – 473: I am surprised that only one soil was only collected in one location per plot. Soil carbon is notoriously heterogeneous in space. How was within-plot variability assessed?

General: There is a distracting overuse of the transition word, “nonetheless,” throughout the manuscript.

Figure 2. Overall, a busy but nice visualization of a lot of data. Please specify in the legend that the values are forest mean \pm se.

Response to Reviewers

Editor

> All editorial requests from the editors in the documents 133706_1_attach_8_354.docx and 133706_1_attach_9_354.pdf have been addressed.

Reviewer 2

Thank you for the very responsive revisions that have significantly improved the manuscript.

> We appreciate these comments. Thank you.

I have a few more specific comments that should be addressed before publication. In addition, I remind the authors that not all reviewers are male – so the use of “he” is not recommended in responses to referees

> We apologize for our clumsy use of “he” to refer to reviewers. We will pay attention in future submission to use gender-inclusive language.

P3L40. More clear: “a decade after initial conversion” instead of “after a decade”.

> Changed as suggested.

P3L40. Watch your tense here and throughout, it seems that the tense switches between past and present.

> Done as suggested. Present tense has been used in the abstract to fulfil the requirements of Nature communications.

P3L41. It would be great if you could specify the efficiency differences (or absolute efficiency) of the three production systems in the abstract.

> This sentence refers to two different efficiencies. We would have to include at least 6 numbers. Moreover, the range of efficiencies is more informative than the mean, which would require to add 12 numbers (see Table 1). Because of the word limit, we think it is more important to specify the kind of efficiency we are speaking of and mentioning that oil palm was the most efficient for both than removing another result or the explanation of which efficiency were assessed.

P3L44. In the interest of acknowledging uncertainty, consider “[could] compromise the ability of...”. Also may want to provide a couple of examples of regulating ecosystem services for clarity.

> Done as suggested. It now reads: *“The imbalanced sharing of NPP between short-term human needs and maintenance of long-term ecosystem functions could compromise the ability of plantations to provide ecosystem services regulating climate, soil fertility, water and nutrient cycles.”*

P4L57-58. “37% of them” should be “with 37% of total area”

> Replaced.

P4L71. Should be “deliver” not “delivers”

> Corrected.

P4L72-74. This sentence is not clear. Please more clearly explain how a high percent of biomass harvest affects energy flow.

> Adjusted according to suggestion: *“Additionally, a high proportion of harvested biomass*

reduces the C and energy available for heterotrophic organisms in the plantation's food web, thereby limiting ecosystem services supporting agricultural production."

P6L117. "with 102-131" is unclear – is this how much carbon was lost? What does the range represent?

> Adjusted according to suggestion: *"As expected, the largest C losses among all C pools occurred in aboveground biomass (AGB), reaching 102-131 Mg C ha⁻¹ depending on the plantation type and corresponding to 73-88% of total C losses."*

P6L119. From reading the methods I see that this is SE. Please note that this is standard error here (the first time presented), as well.

> Done as suggested.

P6L120. Should be "rubber, rubber [and] oil palm plantations" to match syntax of carbon stocks presented previously. Also, are these values time-averaged or for one point in time? Should be clear here even if this information is provided in the methods.

> Corrected. Sentence modified to: *"C losses together at the time of measurement reached..."*

P7L123. +2.7% - percent of what?

> Compared to the previously published AGB stocks estimates.

P7L125. Do the carbon figures presented above use this updated allometric equation or not? If not, why not? Not clear.

> Yes.

P7L126. What differences? Are these non-significant differences based on the Khasanah et al. equations or something else?

> Differences between both estimates (old vs. improved allometric equations).

The text for the last 3 comments reads now: *"The estimated aboveground biomass stocks presented in this synthesis and updated from published data²⁹ using more recent allometric equations^{27,28} did not significantly change for rainforest (+2.7%), jungle rubber (+5.5%) and rubber (+0.3%) plantations as compared to previously published AGB stocks."*

P7L127-132. This section about belowground C is confusing. I think there needs to be a bit more background about what went into these measurements and how they were done. Even though this is presented in the Methods section, short summaries of methods in the Results section will help ground your reader and illustrate how you generated the results.

> We believe the confusion arose because we used different names for the pools to shorten the text, e.g. living root biomass instead of coarse roots and fine roots. We changed back to mention the pools as they appear in figures and tables (coarse roots, fine roots, dead roots and soil organic carbon) and mentioned again that they were measured down to 50 cm depth.

P7L133. (16-28 Mg C ha⁻¹) – is this C loss?

> Yes. We have modified the text to make it clearer: *"The largest belowground losses occurred in roots biomass (16-28 Mg C ha⁻¹ lost depending on land use)."*

P7L139-140. "relative inertia of SOC stocks to land use change" does not make sense.

Consider: "because soil carbon stocks respond more slowly to changes in land use compared to aboveground carbon stocks" or similar.

> Replaced as suggested.

P7L150. Please define in the Results section the threshold you are using to determine significance – if there is room it would be great to report p-values.

> We have added (mean ± standard error, threshold for significance: P-values < 0.05) the first value mentioned in the Results section. It is additionally specified at the end of the methods section that “*If not specified, all discussed differences are significant at a P-value < 0.05.*”. All F- and P- values are reported in the supplementary Table 1. We believe adding F- and P-values to the text would strongly decrease its readability.

P10L191. “previously reported” is confusing – do you mean previously reported in this manuscript, or previously reported in another report? If you mean this manuscript, consider “described above” to clarify.

> We have modified it and included it in the suggestion of Reviewer 3.

P13L277. HCS has not yet been defined as an acronym – please define the first time it is presented. More importantly, the HCSA no longer uses specific carbon thresholds but is more focused on forest structure. Please revise based on the new HCSA Toolkit.

> The reviewer is right to mention that the criteria is now depending on land-cover and not anymore on absolute stocks. Nonetheless, the 75 Mg C ha⁻¹ is still the mentioned in RSPO when farmers want to use the HCS methodology to assess the impact of their new planting and this threshold is used in fundamental research. Accordingly, we have changed the sentence as follows: “*These stocks are also much lower than the threshold of 75 Mg C ha⁻¹ initially defined by the High Carbon Stock (HCS) Approach under which a land converted into plantations would meet the C neutrality criteria³⁸ and still adopted by certification bodies such as RSPO²⁵ or in scientific publications³⁹.*”

P14L287-289. This first sentence is confusing. If calculations just use AGB, and AGB decreases, plantations would be considered C sources, right? Please clarify.

> AGB cannot decrease in plantations having a normal health condition since oil palms constantly grow during the plantation cycle. Therefore, the AGB pool only increases until plantations are replanted. We have added “continues” C accumulation to make it clearer.

P14L289-291. Please revise to be clear that this study does not provide this answer because of the unknown heterotrophic soil respiration fraction. Notably, is probably okay to use “our” and “we” to clarify what was done in this study. As written, it remains less than clear that this was the C budgeting approach taken by these authors. Instead, it sounds like a general approach.

> Done as suggested.

P14L301. “because it implies” – what implies? Revise this sentence for clarity in causality.

> This sentence has been removed because it was redundant with the previous sentence.

P15L302. “Monocultures [are] subject to”??

> The sentence has been modified to state clearly that we were refereeing to a previous study on our research sites.

P16L345. “Oil palm cultivation led to [the] smallest...”

> Corrected.

P17L360. “NPP decreases in the”

> Corrected.

P18L378. “A previous research” is incorrect English grammar here and throughout. Should just be “Previous research”

> Corrected throughout the discussion.

P18L380-382. I don't understand this connection. Is fine root biomass associated with greater water and nutrient demand? You might want to make this clear, instead of assuming that your readers know this.

> Corrected as follows: “*Despite having lower biomass, oil palms tended to invest more in fine roots biomass than plants in rainforest (Fig. 2), suggesting that such high NPP cannot be sustained without elevated water and nutrient consumption by fine roots.*”

P19L400-401. Most if not all intensified systems rely on external inputs. This argument seems to be beyond the scope of this article. Consider revising to focus more specifically on humid tropical plantation systems.

> We did not intend to say that intensified systems rely on external input but that more external input might be necessary to maintain current productivity if ecosystem degradation become too advanced, a potential risk in humid tropical plantation systems supported by our observations. We have modified the sentence to make it clearer as follows: “*Land-use intensification should not reach the threshold where increasing external inputs become necessary to compensate for decreasing resource recycling by internal ecosystem processes⁵⁹, thereby entering a feedback loop of intensification to maintain current land-use productivity, rather than to increase it.*”

P26L553-555. “Differences...” – this is not a complete sentence.

> Corrected

Figure 1. Consider revising so that the figure is colorblind safe – this version will be a problem for people with red-green colorblindness.

> We have modified Figures 1 and 2 accordingly.

Figure 2 (L747). Is the top central figure just a legend? Not clear.

> We have added “*legend*” to avoid confusion.

Reviewer 3

All text corrections in red from the .pdf document provided by Reviewer 3 were done.

Comments in the document are addressed below:

Not clear to what you are referring here? It seems that you are listing all the pools that were measured... Keeps only 2-3 main contributors to the PCA.

Title: down your study as proposed by reviewer 1 and precise where this study occurred. C stocks in African rainforest would be very different...

"Carbon costs and benefits of Indonesian rainforests conversion"

> The title has been modified as suggested.

L 36: What do you mean by plantations of increasing intensity? I suggest rephrasing.

> Rephrased as: “*plantations of increasing management intensity.*”

L 42: Unbalanced sharing of NPP? What do you mean ?

> We have changed unbalanced to imbalanced to stress the inequality of the sharing. See response on comment for L. 175.

l.52 and conflicts with other land uses, especially traditional ones made by local communities.

> We implicitly include traditional land use in food production. We believe it is not necessary to specify explicitly all conflicts. In that case, we would need to also address, e.g. social conflicts about land tenure.

L. 133: Nothing is said on which study/allometry is used to estimate coarse root biomass. Are you referring to: Niiyama et al. 2010?

> Yes, this is the allometry used for forest trees. This information was only available in the original paper of Kotowska. We have now added the references for coarse root estimates in the methods section.

l. 175: I don't understand what you mean by unbalanced share of NPP... do you mean that there are more C that is harvested than stored in the ecosystem? Do you refer to Figure 1? Please clarify.

> We mean that a high proportion of the total NPP is harvested and only a small proportion remains available in the ecosystem. This is an imbalance because it affects ecosystem functioning (see discussion). We have changed the sentence as follows: *“The high productivity of oil palms associated with the removal of a large proportion of their production out of the plantations exacerbated the imbalance share of NPP between human and ecosystem benefits compared to rubber plantations.”*

l. 247 Not clear to what you are referring here? It seems that you are listing all the pools that were measured... Keeps only 2-3 main contributors to the PCA.

> Following the suggestion of the Reviewer we have removed this detailed information from the text since it is given in figures, and the previous sentence already explains the main contributors.

L. 545: The correct citation is: (R Core Team 2017)

R Core Team (2017). R: A language and environment for statistical computing. R Foundation for Statistical Computing, Vienna, Austria. URL <https://www.R-project.org/>.

> We have modified the text but we have not included the reference in the reference list. Our logic is that for instance, we do not add a reference to a company website when using an analytical instrument but we are ready to do it if required.

L. 556: Were the data normalized before partitioning? This is vital if you want to do a sound analysis.

> Yes. They were standardized (centered and divided by standard deviation). We have modified the sentence to make it explicit.

Reviewer 4

The authors present a synthesis of data (mostly previously published) in the form of a carbon budget to evaluate the impacts of rainforest conversion to rubber tree and oil palm plantations. Overall, the authors have done a great job addressing the concerns of the previous reviewers' comments. The revised manuscript is well-written and appropriate for

Nature Communications. The authors seem to have refined previously confusing text in the introduction and methods, and clarified the novel aspects of their work. Figures 1, 2, and 3, in particular, are good visualizations of the conceptual framework and underlying data.

We appreciate these comments. Thank you.

Ln 37 & Ln 386 - 388: I don't think it is necessary to refer to "multidisciplinary" datasets. Ultimately, this work is an ecosystem carbon budget. There are multiple data components (e.g. aboveground, belowground, soil respiration, etc), but I would not characterize that as multidisciplinary datasets. NEP is not a multidisciplinary approach.

> We have removed "Multidisciplinary" in the abstract and have modified "*multidisciplinary approach*" to "*comprehensive dataset*" in the conclusions.

Ln 46 – 50: Agricultural expansion and intensification are two different phenomenon but are used interchangeably here.

> We have now included both expansion and intensification

Ln 75 – 86: This paragraph is a nice discussion of an important gap in our understanding and accounting of land use effects on carbon, soil organic carbon. The discussion (e.g. Ln 315 – 322) circles back to this concept nicely.

> Thank you

Ln 98 – 104: A little more methodological detail would be useful. For example, what were the soil depth intervals, and how many times were soils analyzed for carbon? How often were soil CO₂ and CH₄ flux measurements made?

> To avoid a long enumeration of the multiple technics used to collect the data presented in the manuscript, we have limited the information on the pools and fluxes that were measured, the frequency of measurements and the number of replicates. As suggested, we have added the frequency of gases measurements. The total depth of the considered SOC stocks is mentioned in the previous sentence (50 cm). Since the depth interval is an important but not fundamental information, we have added further information (the names of soil horizons) in the method section. Soil samples were not taken at fixed depth interval but per horizon, which can slightly change between sites. Samples were measured only once for C content since it is not necessary to do technical replicates for this well-established technique. Typically, it consisted in 4 measurements (1 per horizons) down to 50 cm but could vary between 3 and 5 because of the depth variability between sites.

Ln 121 – 125: It is more accurate to say "Aboveground biomass stock estimates..." The updated allometric equation did not change the stock itself, but rather the authors' calculated estimate of the stock.

> Corrected.

Ln 129 – 132: This is a confusing sentence. Did the plantations in the loamy Acrisol regions experience SOC loss compared to the other region with a clayey Acrisols, or compared to the rainforest? If soil type is a key variable, why was soil type ignored in the statistical analysis?

> We have now specified that it is the comparison with rainforest. Our aim was to analyze the whole dataset in the same way. The RDA showed that the factor region (having different soil texture) explained only 5% of the overall dataset variability while land-use accounted for 47%. Therefore, we did not add the factor region in our model. We agree with the reviewer that in the case of SOC, overlooking this factor has limited the ability to show differences between land-use types in this pool. However, a detailed analysis was performed in the

original article showing an effect of plantations on SOC in the top soil in only one region. We preferred mentioning this in the results, referring to the original article and keeping a consistent way to analyze all pools/fluxes and their sum throughout this synthesis.

Ln 168 – 170: How are the dead oil palm frond piles managed? Are they left to passively decompose or are they composted aerobically?

> Dead palm fronds are not composted, just left decomposing passively (which might be an explanation why so little OM from fronds is stabilized in SOC)

Ln 360: remove the word “in”

> Done

Ln 421 – 422: Do you mean “...had not been fertilized since the year of plots establishment”?

> We meant during the time measurement were taken. The text has been modified to: “...,rubber monoculture had not been fertilized the year measurements took place,...”.

Ln 472 – 473: I am surprised that only one soil was only collected in one location per plot. Soil carbon is notoriously heterogeneous in space. How was within-plot variability assessed?

> We acknowledge that within-plot variability was not assessed by this sampling, which could have reduced the power of our design to show differences between land use. Nonetheless, another study on the same 32 sites (ref 21: Allen et al. 2016) showed using a variance partitioning analysis on 10 sampling point for each site that soil parameters variance was largely dominated by the variance among replicate sites and regions. The variance within site accounted for less than 10% of the total variance.

General: There is a distracting overuse of the transition word, “nonetheless,” throughout the manuscript.

> We have removed or modified the conjunction when appropriate.

Figure 2. Overall, a busy but nice visualization of a lot of data. Please specify in the legend that the values are forest mean +/- se.

> This information was present in the legend. We hope that it is now more visible after the modifications of the legend and we have added the information in the figure caption as well.